**Investigation**

# On the estimation of genome-average recombination rates

Julien Y. Dutheil [ID] *

Max Planck Institute for Evolutionary Biology, August-Thienemann-Str. 2, Plön 24306, Germany

*Corresponding author: Max Planck Institute for Evolutionary Biology, August-Thienemann-Str. 2, Plön 24306, Germany. Email: dutheil@evolbio.mpg.de

The rate at which recombination events occur in a population is an indicator of its effective population size and the organism's reproduction mode. It determines the extent of linkage disequilibrium along the genome and, thereby, the efficacy of both purifying and positive selection. The population recombination rate can be inferred using models of genome evolution in populations. Classic methods based on the patterns of linkage disequilibrium provide the most accurate estimates, providing large sample sizes are used and the demography of the population is properly accounted for. Here, the capacity of approaches based on the sequentially Markov coalescent (SMC) to infer the genome-average recombination rate from as little as a single diploid genome is examined. SMC approaches provide highly accurate estimates even in the presence of changing population sizes, providing that (1) within genome heterogeneity is accounted for and (2) classic maximum-likelihood optimization algorithms are employed to fit the model. SMC-based estimates proved sensitive to gene conversion, leading to an overestimation of the recombination rate if conversion events are frequent. Conversely, methods based on the correlation of heterozygosity succeed in disentangling the rate of crossing over from that of gene conversion events, but only when the population size is constant and the recombination landscape homogeneous. These results call for a convergence of these two methods to obtain accurate and comparable estimates of recombination rates between populations.

Keywords: recombination; sequentially Markov coalescent; gene conversion; demography

## Introduction

Recombination is a fundamental process impacting the segregation of alleles in populations and, consequently, a driver of genetic diversity between populations and within genomes. The average recombination rate was shown to vary extensively between individuals, populations, and species (Stapley *et al.* 2017). The amount of recombination occurring in a population is an indicator of the species' reproduction mode, providing insights into the frequency of sexual reproduction in natural populations (e.g. in plankton species Rengefors *et al.* 2017). The recombination rate also determines how efficiently populations get rid of deleterious mutations and fix advantageous ones, with implications for conservation biology (Theissinger *et al.* 2023) and breeding strategies (Epstein *et al.* 2023).

Recombination rates can be inferred either directly or indirectly (Peñalba and Wolf 2020). Direct assessments provide measures of the contemporary molecular recombination rate and involve experimentally demanding methods such as multiple crossing and genotyping. Indirect methods involve genome resequencing in combination with variant calling pipelines, and (1) require comparatively less laboratory work, (2) are also amenable to organisms that cannot be grown in the laboratory. These methods rely on the patterns that recombination leaves on the distribution of variants within populations; they infer the so-called population recombination rate, or scaled recombination rate, noted $\rho$, equal to $2 \cdot x \cdot Ne \cdot r$, where $Ne$ is the effective population size, $x$ the ploidy of the individuals serving as unit count for $Ne$, and $r$ is the "molecular" recombination rate, the number of recombination events

per base pair per generation. As opposed to direct methods, indirect methods provide a historical recombination rate, averaged over multiple individuals and generations.

The main signature of recombination in the sequence data used to infer $\rho$ is linkage disequilibrium (LD) between variants. *LDhat* was a pioneering method for reconstructing fine-scale recombination maps from population genomics data (McVean *et al.* 2004). The resulting inferred rate is population- and sex-averaged over time and individuals. Yet, this method unraveled the variation of this rate along genomes with an unprecedented resolution. *LDhat*'s methodology served as the basis for two decades of developments, accounting notably for demographic history (Kamm *et al.* 2016) and improving scalability to increasingly larger datasets (Spence and Song 2019).

In parallel to the development of methods to infer $\rho$, models based on the sequentially Markov coalescent (SMC) have unleashed the power of full genome data for demography inference (Spence *et al.* 2018). The SMC is an approximation of the sequential coalescent with recombination (Wiuf and Hein 1999). SMC models have a recombination rate parameter, which can be jointly estimated with demographic parameters in inference methods like PSMC (Li and Durbin 2011) and MSMC2 (Schiffels and Wang 2020). Because they ignore certain classes of recombination events (Marjoram and Wall 2006), SMC models are expected to underestimate $\rho$. Furthermore, most SMC models use a single parameter for the entire analyzed genome, an assumption at odds with many empirical recombination landscapes, which display skewed rate distributions. Conversely, the integrative sequentially Markov coalescent iSMC method accounts for the heterogeneity

of the recombination rate by assuming a prior distribution and an auto-correlation process along the genome (Barroso *et al.* 2019). In this model, the genome-wide $\rho$ is estimated as a hyper-parameter (the mean of the prior distribution). While SMC-based approaches do not reach a fine-scale resolution near that of LD-based methods, they permit the inference of $\rho$ with as little as one unphased diploid genome.

Other approaches make use of LD properties to extract information about $\rho$ with small sample sizes. Just like SMC-based methods, methods relying on the correlation of heterozygosity provide estimates of genome-average population recombination rates (Haubold *et al.* 2010). However, such methods assume a homogeneous recombination rate along the analyzed sequences and do not permit the inference of local variation. However, a recent extension, *heRho*, permits disentangling the crossing-over (CO) rate from the rate of gene conversion (GC) events (Setter *et al.* 2022). The performance of other LD-based and SMC-based methods in the presence of GC is yet to be characterized.

We here compare the performance of distinct population genomics methods to infer the genome average $\rho$, for the purpose of applying them to a broad range of (nonmodel) organisms. This assessment is made with respect to sample size (with a focus on small samples), the shape of the recombination landscape (homogeneous/variable/with hotspots), the demography of the population, and the presence of GC in addition to CO events. Finally, I also compare the methods' behavior when they are applied to data without recombination.

## Materials and methods
### Simulations

Simulated datasets of 10 Mb were generated using the *msprime* package version 1.2.0 (Baumdicker *et al.* 2022). All simulations were performed with a per-generation mutation rate of 1.25e−8 bp$^{-1}$ and an effective population size of 100,000 diploid individuals. Ten replicates were generated in each scenario. The resulting datasets were exported in the variant call format (VCF), the resulting VCF files serving as input for the various inference methods. The analyzed scenarios varied according to the considered recombination landscape and demographic history. Several average $\rho$ were considered: 0, 0.1, 0.5, 1, and 5 times 1.5e−8 M.bp$^{-1}$, corresponding to $\rho = 4 \cdot Ne \cdot r = 0$, 0.0006, 0.003, 0.006, 0.03.

#### *Variable recombination landscape*
Random recombination maps were generated by drawing segments with lengths taken from a geometric distribution of mean 10 kb. The recombination rate in each segment was drawn randomly from an exponential distribution with a mean of 1.0. The generated recombination maps were then multiplied by the average recombination rate.

#### *deCODE recombination landscape*
As an alternative recombination map, the sex-averaged deCODE recombination map was used (Kong *et al.* 2002). The first 10,000 10 kb windows of Human chromosome 1 were standardized in order to have a mean of 1.0 before being multiplied by the specified averaged recombination rate.

#### *Recombination landscape with hotspots*
Hotspots were simulated using a model similar to that of the variable recombination landscape. Regions of uniform rates were generated, with a rate sampled from an exponential distribution of mean 1.0 and a length sampled from a geometric distribution

with a mean of 50 kb. As in the variable recombination model, the "background" rate of these regions was then multiplied by the specified average recombination rate. Hotspots with a recombination rate of $70 \times 1.25$e−8 bp$^{-1}$ were inserted between each region, with a breadth sampled from a distribution uniform between 0.5 and 2 kb. In this model, the genome average recombination rate is a weighted average of the background rate, $\rho_{\text{bg}}$, and the recombination rate in the hotspot regions, $\rho_{\text{hs}} = 70 \times 1.25$e−8 bp$^{-1}$:

$$\rho = (1 - \lambda) \cdot \rho_{\text{bg}} + \lambda \cdot \rho_{\text{hs}}, \tag{1}$$

where $\lambda$ is the proportion of the simulated sequence located within hotspots.

#### *Gene conversion*
GC events were added, as implemented in the *msprime* simulator (Baumdicker *et al.* 2022). The GC rate $\rho_{\text{GC}}$ was provided as a proportion $\gamma$ of the total population recombination rate $\rho$, so that $\rho_{\text{GC}} = \gamma \cdot \rho$ and the rate of CO events $\rho_{\text{CO}} = (1 - \gamma) \cdot \rho$. GC track lengths are taken randomly from a geometric distribution of mean 300 bp. Additional simulations were run with a track length of 2 kb for comparison.

#### *Nonconstant demographic scenarios*
We considered three scenarios departing from a constant population size. In the first scenario (referred to as "population decline" in the following), an ancestral population size of $Ne = 100,000$ exponentially decreases to $Ne = 1,000$ at present time, starting 1,000 generations before present. This population decline had little impact on the resulting genetic diversity ($\theta = 0.0046$ vs $\theta = 0.0049$, when a flat recombination landscape is used). In a second scenario (referred to as "recent population growth" in the following), we considered a scenario where an ancestral population of size 50,000 increased exponentially to a size of 200,000, starting 1,000 generations ago. We note that the genetic diversity in this growth scenario is roughly half that of the constant or decreasing population size scenarios ($\theta = 0.0026$), owing to the starting population size being 50,000 instead of 100,000 in the two other scenarios. Finally, we considered an "ancient population growth" scenario, where the population size increases from 10,000 to 200,000, starting 100,000 generations ago, resulting in a $\theta = 0.0018$.

Importantly, under a nonconstant population size scenario, $\rho = 4 \cdot Ne \cdot r$ varies in time, even if $r$ is constant. The inferred $\rho$ is, therefore, a time average of the population recombination rate. When plotting the simulated population $\rho$, in the absence of selection, the average $Ne$ is estimated using the formula $Ne = \pi / (4 \cdot u)$, where $\pi$ is the average pairwise heterozygosity of the simulated dataset (computed using the *vcftools* (Danecek *et al.* 2011)) and $u = 1.25$e−8 is the mutation rate.

## Inference of genome-average population recombination rate

For each simulated scenario—a combination of a recombination landscape, demographic scenario, average recombination rate, and sample size—we inferred the genome-average population recombination rate with distinct methods for 10 independent replicates.

### *The multiple sequentially Markov coalescent, version 2*
Individual genome data were extracted from the simulated VCF files using the *vcftools* (Danecek *et al.* 2011) and then converted into multiple sequentially Markov coalescent (*MSMC*) input files

using the provided python script *generate_multihetsep.py*. MSMC2 (Schiffels and Wang 2020) was run with default parameters and with a restricted epoch scheme with only 16 intervals, using the argument `-p 1*2+11*1+1*3`. A model with a constant population size and 30 time intervals (`-p 1*30`) was also fitted for comparison. Under the homogeneous rate scenario, MSMC2 was also run with 40 iterations instead of the default 20 (argument `-i 40`), and with an initial $\rho/\theta = 4$ instead of the default 0.5 (argument `-r 4`). The resulting population recombination rate was read from the last line of the output `*.loop.txt` file. If the program failed to converge and stopped before the 20 (resp. 40) iterations, the resulting estimate was recorded as missing data.

### The integrative sequentially Markov coalescent

*iSMC* (Barroso *et al.* 2019) was run on the simulated VCF files with a two-knots spline model, 40 time intervals, and a precision of 0.1 on the log-likelihood. The method was referred to as *iSMC* when a homogeneous model was considered. We further considered a model with a five-class discrete gamma model of recombination, which we here note *rhoSMC*. Under this model, the genome-average recombination rate $\rho$ is the mean of the gamma prior distribution of rates and is estimated together with other parameters using a maximum likelihood approach. The *rhoSMC* model further permits to use an empirical Bayesian approach to estimate site-specific recombination rates by computing the mean of each recombination class weighted by their site-specific posterior probabilities, computed using the backward algorithm (Dutheil 2021). A posterior estimate of the genome-average population recombination rate is then computed as the mean over all site-specific posterior estimates.

### LDhat

The simulated VCF files were converted to *LDhat* input files using the *vcftools* (Danecek *et al.* 2011). No filtering on the minimum allele frequency was performed, and all SNPs were kept. The interval program from the *LDhat* package was run with options `-its 10,000,000 -bpen 5 -samp 5,000` (10 million iterations sampled every 5,000 generations, with a penalty of 5), as suggested in the user manual. Average $\rho$ were computed using the *stats* program from the package, using option `-burn 20` to discard the first 20 samples, corresponding to a burn-in of $20 \times 5,000 = 100,000$ iterations. The interval program being very slow to converge when the simulated $\rho$ is zero, it was not run in these conditions for a sample size of 50 diploids. The corresponding estimates were recorded as missing data. The genome-average estimates were calculated using the average of all inter-loci recombination rates, weighted by the distance between the corresponding SNPs.

### Pyrho

The *pyrho* software (version 0.1.6) was run using the recommended parameters (Spence and Song 2019). Likelihood tables were generated for each demography scenario used when simulating the data. In the case of nonconstant population sizes, 20 time intervals of piece-wise constant population sizes were considered. For instance, in the "population decline" scenario, the population sizes $N_t$ of each interval $t$ were computed using the formula

$$N_t = \text{round}(1e5 \cdot \exp(g \cdot (1e3 - t))), \tag{2}$$

where the growth rate $g = -\frac{\log(1e5/1e3)}{1e3}$ (exponential decline for a population of 100,000 diploids 1,000 generations ago, to a size of

1,000 diploids at present). Similar calculations were performed in the population growth cases.

The impact of hyper-parameter choice was assessed using a grid of three block penalties and three window sizes. We found that the default parameters offered a good compromise across all criteria and kept them. The genome-average estimates were calculated using the average of all inter-loci recombination rates, weighted by the distance between the corresponding SNPs.

### heRho

VCF files were converted to *heRho* input files using the provided *heRho_tally_pairwise_counts_vcf.py* script. *heRho* was then run using default parameters (Setter *et al.* 2022). We reported two recombination rates from the *heRho* output: the CO rate (kappa parameter) and the total recombination rate (kappa + omega), which includes GC events.

## Statistical analysis

The estimated $\rho$ values were plotted against the simulated CO rates. All analyses were done in R (version 4.1.0) (R Core Team 2021) with the *ggplot2* package (version 3.4.2) (Wickham 2016).

## Results

The comparison benchmark used in all the following consists of (1) simulating individual sequences according to a demographic scenario and recombination landscape, (2) using the generated data to infer the average genome population recombination rate (hereby noted $\rho$) using several population genomics methods, and (3) comparing the resulting estimate to the simulated $\rho$ value. The simulated sequences always consist of a unique chromosome of size 10 Mb and ten replicates. The mutation rate was set to that of humans, $1.25e-8$ $\text{bp}^{-1}$ and kept constant in all analyses. The combination of the mutation rate parameter and chromosome length ensures that there is enough diversity to fit models while keeping the computational time feasible over multiple replicates. The 10 Mb chromosome length also permits assessing the methods' accuracy on small genomes such as the ones of several microorganisms. The number of individuals used for inference, the demographic scenario, recombination landscape, and inference method vary in each experiment. The range of average population recombination rates ($\rho \in [0, 0.03]$) and landscapes (flat, heterogeneous, deCODE map, with hotspots) covers a broad range of biological conditions. Four simple demographic scenarios are tested: constant population size, population decline, recent, and ancient population growth. The focus of this work is not to assess the ability of the methods to infer demography but rather to evaluate the robustness of recombination inference to deviations from the assumption of a constant population size.

## Ideal scenario: flat recombination landscape and constant population size

We first considered an ideal scenario where the data is generated using a model very close to the inference model. The generative model consists of a standard coalescent with recombination, with constant population size in time and along the genome (no selection), and a flat recombination landscape (homogeneous recombination). We compared two SMC-based methods: the multiple sequentially coalescent (version 2), MSMC (Schiffels and Wang 2020), and the integrative sequentially Markov coalescent, iSMC (Barroso *et al.* 2019). For this comparison, iSMC was used with a single recombination class so that its model is identical to MSMC. As a comparison, two linkage disequilibrium (LD) based

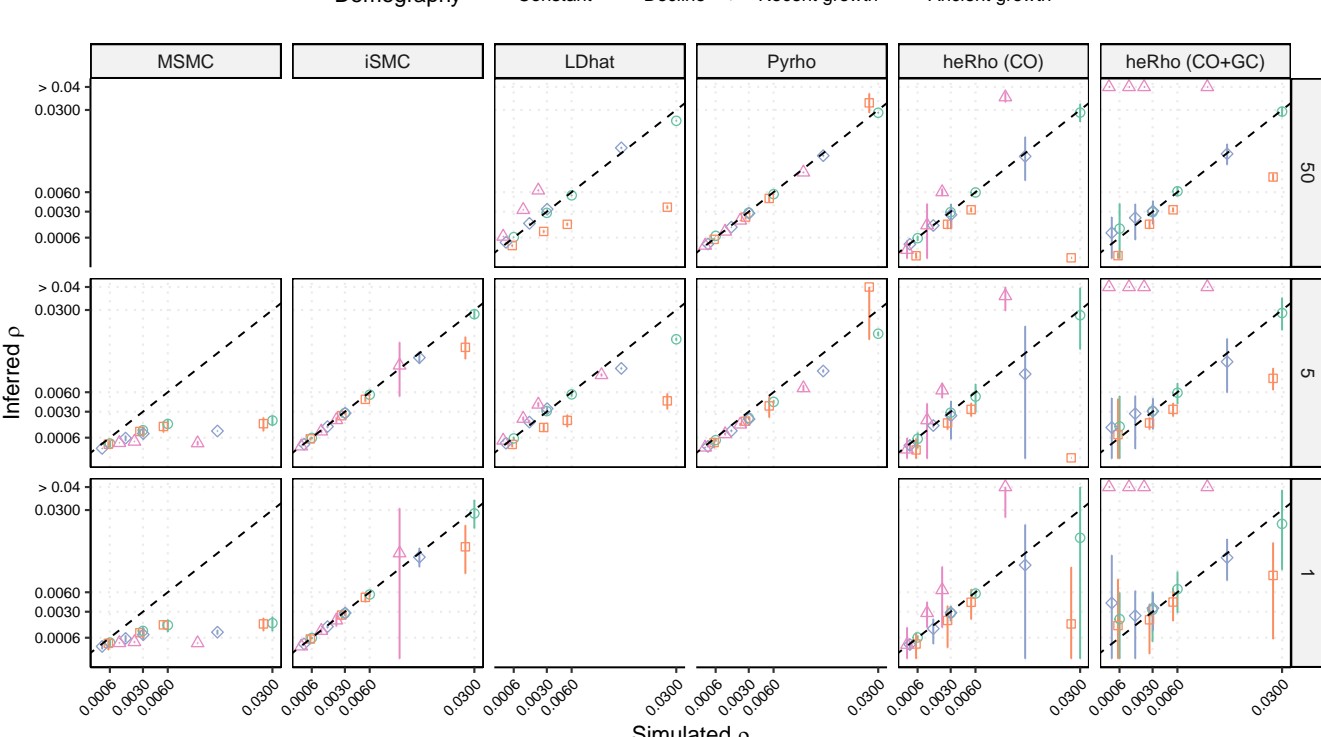

**Fig. 1.** Inference of the genome-wide population recombination rate under a flat recombination landscape. The inferred rate is plotted on the y-axis for distinct methods (column facets) and sample size (row facets, number of diploid individuals) as a function of the simulated rate on the x-axis. The points represent the average over all 10 replicates and the error bars the 95% confidence interval of the mean, assuming normality (mean ± 1.96 · standard deviation). As the *heRho* method estimates the CO and GC rates, both the CO and total (CO+GC) rates are reported. Each point represents the mean value over ten replicates, with the corresponding 95% confidence interval (± 1.96 · standard deviation). Both axes are plotted using a square-root scale, and the 1:1 diagonal is plotted as a dashed line. SMC methods cannot be run on large sample sizes (such as n = 50, while LD-based methods cannot be run on single diploid genomes (n = 1).

methods were also included: *LDhat* (McVean *et al.* 2004) and *Pyrho* (Spence and Song 2019), a more recent re-implementation of the *LDhat* method. Lastly, we included the *heRho* method (Setter *et al.* 2022), which estimates ρ from the distance patterns of heterozygous sites.

Under this ideal scenario, most methods recovered ρ with high accuracy (Fig. 1, constant population size scenario). LD-based methods slightly underestimated ρ when the sample size was small (5 diploids) and the recombination rate was high. *heRho* accurately recovered the CO rate but tended to estimate a nonzero rate of GC when the total recombination rate was low. Furthermore, *heRho* estimates had a larger variance than SMC-based methods when the sample size was small.

*MSMC* systematically underestimated ρ values. The difference between *MSMC* and *iSMC* is particularly surprising, given the strong similarity of the underlying model. Besides their independent implementations, the *iSMC* and *MSMC* methods differ in the demographic model specification (skyline model for *MSMC* vs splines for *iSMC*) and the optimization algorithm. The first point is not expected to play any role in the observed differences since the true population size is constant. As a control, we observe that *MSMC* recovered the simulated demography with fairly good accuracy with the exception of the largest recombination rate, where it inferred an initial slight decline followed by a regrowth (Ishigohoka and Liedvogel 2024). The variance between replicates was typically larger in the most recent and most ancient time intervals (Supplementary Fig. 1). Convergence issues arise when the analyzed genomes are too short, and following the

*MSMC* authors' recommendation, we reduced the number of time intervals in the skyline model to improve the performances (Schiffels and Wang 2020). Using half of the default time intervals (see Inference of genome-average population recombination rate in Material and Methods) provided a performance gain, reducing the variance in the inferred population sizes and reducing the number of convergence failures. However, it had no impact on the estimation of ρ (Supplementary Fig. 2). To disentangle the effect of discretization (impacting the number of hidden states in the hidden Markov chain) from that of the number of parameters to estimate, we fitted a model with flat demography using 30 intervals and a single population size parameter. Such a model only improved the inference of ρ when the simulated recombination rate was zero (Supplementary Fig. 2). Doubling the number of iterations in the Baum–Welch algorithm or starting from a different initial value did not improve the recombination estimation (Supplementary Fig. 2).

SMC-based approaches are typically applied to a small number of genomes, possibly as little as a single diploid individual (or two haploid genomes). The estimation of ρ did not show any noticeable difference when a single unphased diploid (two haploid genomes) was used compared to five diploids (Fig. 1). Finally, we note that the ρ/θ ratio, where θ = 4 · Ne · u = 0.005, varied from 0.12 to 6 throughout the simulations, showing that SMC-based approaches can infer the genome-average ρ even in situations where ρ/θ > 1 and some recombination events leave no trace in the data.

In conclusion, we note that under an ideal scenario with flat recombination landscape and constant population size:

1) The SMC permits the genome-average $\rho$ inference with good accuracy and with as little as one diploid individual.
2) A standard maximum likelihood optimization procedure should be used when estimating $\rho$, as the Baum–Welch expectation–maximization algorithm does not permit obtaining accurate estimates.
3) When the simulated recombination rate is high and the sample size is small, the SMC-based estimates are less biased than those of the LD-based methods.
4) The estimation variance is lower for SMC-based estimates than for the heterozygosity-based method.

## Nonconstant population size

The population recombination rate depends on the product of the molecular recombination rate, $r$, and the population size, $Ne$. Assuming that the genome average $r$ is constant over the period since the most recent common ancestor of the sample is a relatively reasonable assumption. Conversely, variation in $Ne$ over this time scale may strongly impact the estimation of $\rho$ (Kamm et al. 2016). We simulated data under three simple demographics featuring exponential decline or growth. In the "population decline" scenario, the population size was set to 100,000, as in the constant population size scenario, but it exponentially decreased starting 1,000 generations ago to reach a population size of 1,000 at present. In the "recent growth" scenario, a population of 50,000 increased exponentially to a size of 200,000 starting 1,000 generations ago. Finally, in the "ancient growth" scenario, the population size exponentially grew from 10,000 to 200,000, starting 100,000 generations ago. While a recent population growth had little impact on the results, population decline led to a strong underestimation of $\rho$ when methods that assume a flat demography (LDhat, heRho) were used (Fig. 1). Conversely, $\rho$ was overestimated by these methods when the growth was ancient. The LD-based method Pyrho provided accurate estimates since the underlying demography is taken into account when computing the likelihood table with the LDpop method (Kamm et al. 2016). iSMC also recovered the simulated $\rho$ as it jointly estimates demography with the recombination landscape, although with a large variance when $\rho$ was high.

In conclusion, we note that under a flat recombination landscape scenario and nonconstant population size:

1) Ignoring population size variation leads to biased estimates of $\rho$.
2) Accounting for demography when computing the likelihood table permits accurate inferences using LD-based methods.
3) The iSMC method accurately infers the genome-average $\rho$ when the demography is not constant, although with large variance when $\rho$ is high.

## Heterogeneous recombination landscape

We used the same setup as in the flat recombination landscape scenario, but the recombination landscape was no longer homogeneous along the chromosome. Segments of uniform recombination rates were randomly generated, their lengths taken from a geometric distribution and their values exponentially distributed (see Simulations in Material and Methods). In addition to the methods tested in the previous case, we added the iSMC method with five recombination classes (noted rhoSMC in the following, Barroso et al. (2019)). The rhoSMC model includes the genome average $\rho$ as a parameter that can be estimated by maximum likelihood (referred to as maximum likelihood estimate, MLE, in the following, similar to the simpler iSMC model. In addition, rhoSMC allows computing a posterior average estimate (referred to as posterior estimate in

the following, see Inference of genome-average population recombination rate in Material and Methods).

Under a constant population size scenario, methods assuming a homogeneous recombination rate underestimated the genome average rate (iSMC and heRho), while LD-based methods, which infer a recombination map, showed the same performance as when the landscape is flat (only slightly under-estimating high $\rho$ values when the sample size was small, Fig. 2). While heRho generally underestimated the CO rate; in this scenario, we note that its total recombination rate estimate was closer to the simulated rate, suggesting that heRho interprets some of the signatures of variable recombination rate as GC events. rhoSMC's estimates were unbiased (Fig. 2); the MLE and posterior estimates were similar, the posterior estimate being slightly superior when the recombination rate was low. We note that the methods' performances were unaffected when simulating under a population size 10 times smaller, resulting in a 10 times lower genetic diversity (Supplementary Fig. 3).

As observed in the flat recombination scenario, not accounting for the history of population size changes impacted the estimation of $\rho$. The average $\rho$ was under or overestimated by inference methods that assume a constant population size (LDhat, heRho). While the tested recent population growth scenario (4-fold increase with reduced genetic diversity) had a mild impact, the population decline (recent, 100-fold decrease with no strong impact on the genetic diversity) and ancient growth (20-fold increase starting 100,000 generations in the past) led to a strong bias (Fig. 2). Notably, the underestimation of iSMC was identical in all four scenarios, suggesting that it stems from the heterogeneous recombination landscape itself rather than the underlying demography. As previously reported (Li and Durbin 2011; Schiffels and Durbin 2014), we observed that MSMC accurately recovered the underlying demography, despite assuming a homogeneous landscape and inaccurately inferring the average $\rho$ (Supplementary Fig. 4), confirming that demography inference per se is robust to the model of recombination landscape.

The auto-correlation model used to generate the recombination map of the simulations was very similar to the recombination rate variation model of rhoSMC, the latter using discrete rate categories instead of continuous recombination rates. To assess the performance of rhoSMC when the recombination landscape is not generated by this simple auto-correlation pattern, the analysis was repeated on a recombination landscape taken from the deCODE human recombination map (see Materials and methods); the results and conclusions proving to be qualitatively similar. We note a higher variance between replicates for the inference with rhoSMC and a larger underestimation with heRho on this landscape, compared to the randomly generated one (Supplementary Fig. 7).

Next, we investigated the performance of rhoSMC—which assumes a heterogeneous recombination landscape—when the simulated landscape is homogeneous. The maximum likelihood estimator appeared to be biased under this scenario, while the posterior average estimate proved to be more robust and recovered the simulated $\rho$ value under demographic scenarios not too far away from a constant population size. The $\rho$ estimate remained strongly biased, however, when the simulated recombination rate was zero or under the ancient growth scenario (Supplementary Fig. 5). This result suggests that some model testing should be done before interpreting the estimated values.

The heterogeneous rhoSMC model (M1) can be compared to the homogeneous iSMC model (M0) by setting the number of recombination categories to 1. The presence of recombination

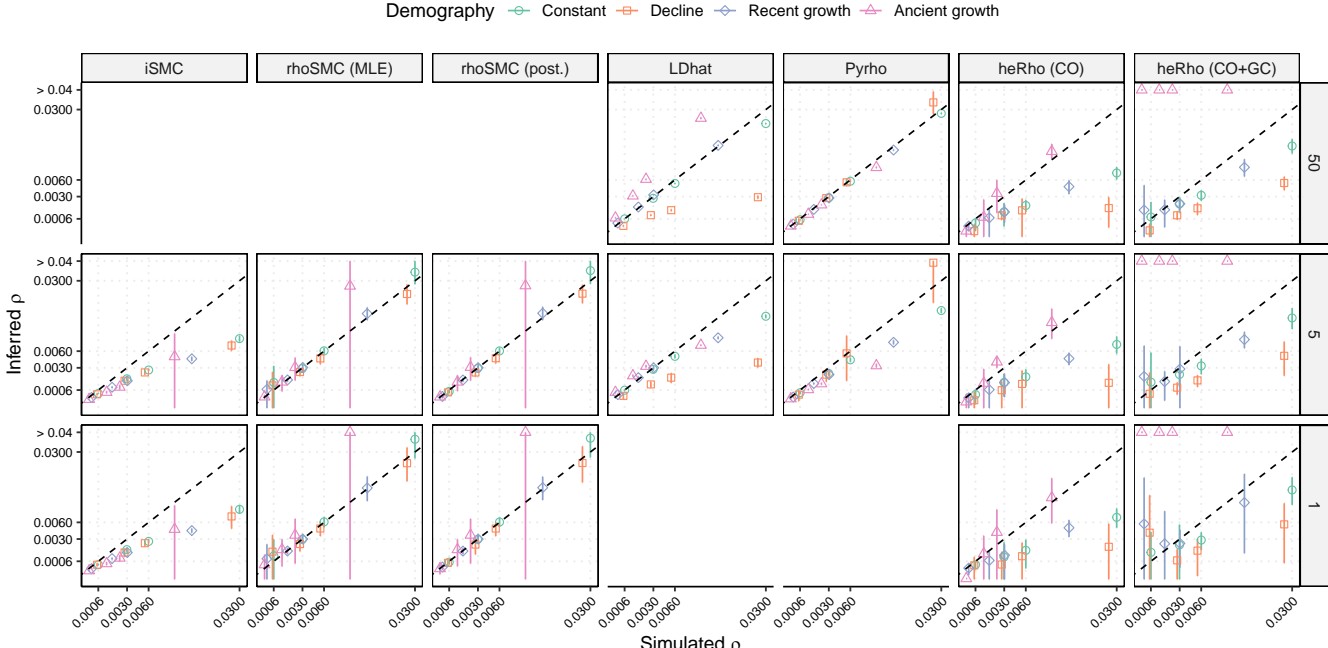

**Fig. 2.** Inference of the genome-wide population recombination rate under a heterogeneous recombination landscape. Legend as in Fig. 1.

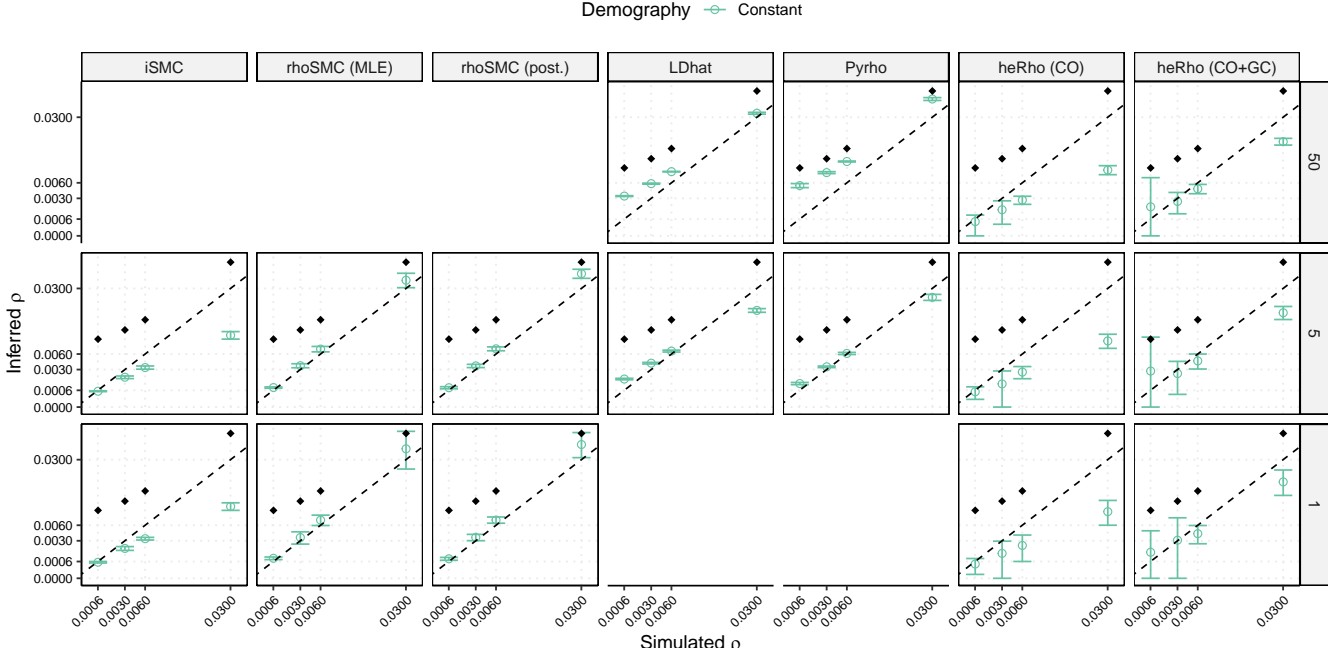

**Fig. 3.** Inference of the genome-wide population recombination rate in the presence of recombination hotspots. Simulations under a constant population size. The x-axis shows the background rate. Circles show the inferred ρ as an average over 10 replicates with 95% confidence intervals. Diamonds show the simulated genome-average ρ values as a weighted average of background and hotspots. Other legends as in Fig. 1.

heterogeneity can be assessed by comparing the two models using Akaike's information criterion (AIC) or a likelihood ratio test (LRT). When computed on a homogeneous dataset, the AIC difference between the two models was roughly distributed between −10 and 10, independently of the recombination rate. However, it reached −1e6 when ρ is 0 and 5 individuals were used and tended to be negative under a decreasing population size scenario (Supplementary Fig. 6a). In comparison, the AIC differences reach −1,000 when applied to a heterogeneous landscape. While AIC seems able to disentangle the two models, the threshold to use

is ambiguous, meaning that simulations tailored for the dataset to analyze are required to make a proper goodness-of-fit test. Similar conclusions were reached when considering the LRT, where lower P-values were obtained on heterogeneous datasets, but P-values lower than 1%, and even 0.01%, were observed with homogeneous datasets (Supplementary Fig. 6b). Lastly, the presence of GC mimics rate heterogeneity, leading to a stronger signal for rejecting M0 when the simulated landscape is homogeneous (Supplementary Fig. 6c and d). We further discuss the effect of GC below (see Gene conversion in Results).

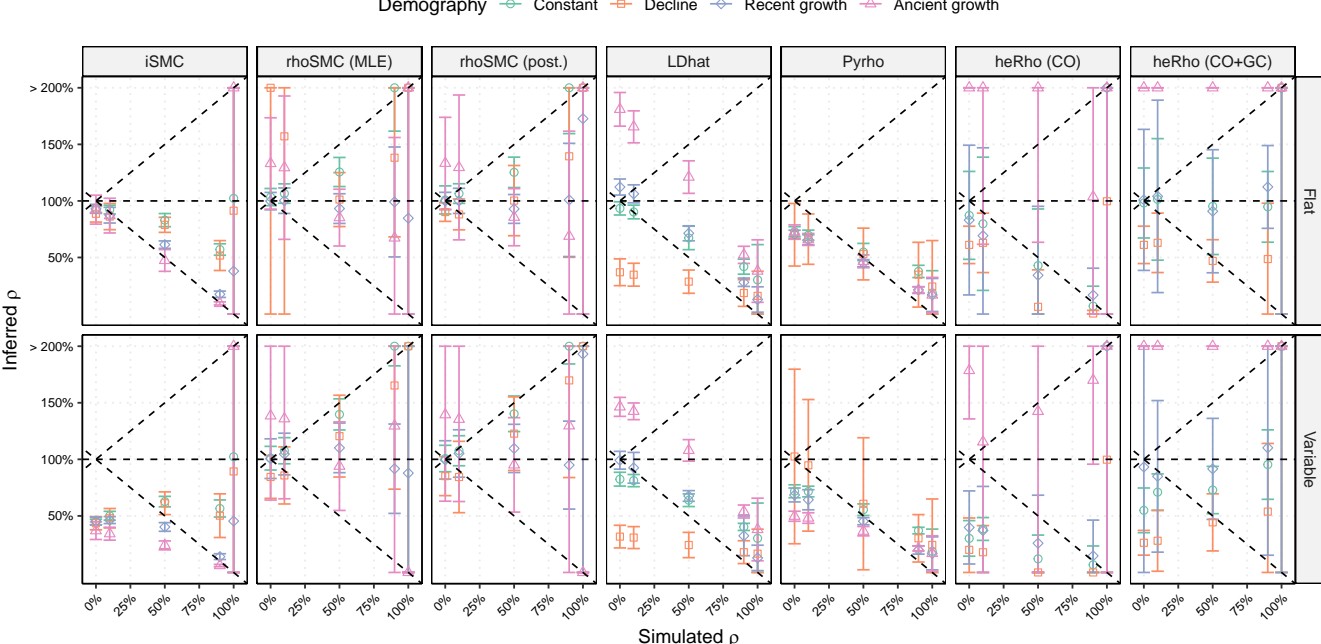

**Fig. 4.** Inference of the genome-wide population recombination rate in the presence of gene conversion (GC). Simulations under a recombination rate of 1.5 cM/bp with various proportions of GC events (x-axis). The y-axis reports the ratio of the inferred $\rho$ to the simulated one as percentages. Facet rows display various scenarios, with a sample size of 5 diploids in all cases. Results for sample sizes of 1 are shown in Supplementary Fig. 8 and 50 in Supplementary Fig. 9. Horizontal dash lines correspond to the simulated relative $\rho$ (100%), and the −1 slope dashed line shows the CO rate as a proportion of the total $\rho$. The +1 slope dashed lines show the theoretical total $\rho$ when GC events count as two recombination events. Methods falling on the $y = −x$ line correctly estimate the CO rate, while methods falling on the $y = 0$ horizontal line correctly estimate the total $\rho$.

We conclude that under a heterogeneous recombination landscape:

1) Assuming a homogeneous recombination landscape in the SMC leads to underestimating the average rate, even when a pure maximum likelihood optimization framework is used.
2) Similarly to *iSMC*, *heRho* underestimates the average $\rho$ when the landscape is heterogeneous.
3) Accounting for the recombination rate variation in the SMC permits an unbiased inference of the average $\rho$, even when the underlying recombination landscape differs from the underlying model of rate variation.
4) The SMC maximum likelihood estimates of the average $\rho$ are biased when a heterogeneous model is used and the simulated landscape is homogeneous. Posterior average estimates show better performance under these conditions.
5) As in the flat-recombination scenario, not accounting for historical population size changes may result in an estimation bias.
6) AIC and LRT model comparison tests do not permit to distinguish SMC models with heterogeneous and homogeneous recombination properly. Tailored simulations are required to derive significance likelihood thresholds.

### Recombination hotspots

In many species, the recombination landscape is shaped by recombination hotspots, short regions with very high recombination rates. Their localized nature and the high number of recombination events make them difficult to infer and require a high-resolution recombination map. To assess the accuracy of the inference of the genome-average $\rho$ in the presence of hotspots, we simulated random regions of 0.5 to 2 kb with $\rho = 0.35$ (on

average 70× the background rate) spaced by uniform regions of 50 kb on average (see Recombination landscape with hotspots in Materials and methods). We observed that all methods underestimated the average $\rho$ under this scenario (Fig. 3). Only LD-based methods captured a partial signal from the recombination hotspots when a large sample size was available. Methods applied to small sample sizes (iSMC and LD-based methods) only inferred the background $\rho$. Given their pairwise nature, SMC-based methods are, in particular, blind to the presence of recombination hotspots.

Conclusions:

1) Only LD-based methods and large sample sizes capture the effect of recombination hotspots.
2) SMC-based methods efficiently measure the background population recombination rate.

### Gene conversion

We assessed the impact of GC events on the genome-average $\rho$ inference. The simulation setup was identical to the above, but a variable proportion of GC events was implemented. We kept the average total population recombination rate (CO + GC events) equal to $\rho = 0.006$ and implemented proportions of GC of 0%, 10%, 50%, 90%, and 100%, with an average track length of 300 bp (see Gene conversion in Materials and methods). While a proportion of 100% GC is unrealistic for eukaryotic genomes, it is analogous to a bacterial recombination process (De Maio and Wilson 2017).

Most tested methods performed poorly when GC was present (Fig. 4). *heRho* was the only method able to infer the total and CO recombination rates properly. We note, however, that this was only the case when the recombination landscape was flat, and the population size was constant, $\rho$ being generally underestimated

otherwise. We note that *heRho*'s estimates were highly overestimated in the presence of an ancient population growth. Under a flat recombination landscape, the *iSMC* method inferred an intermediate rate, providing GC events did not represent 100% of the recombination events. The estimation was, however, affected by demography, with lower rates being inferred in the presence of population growth. The *rhoSMC* method proved highly sensitive to the presence of GC events, resulting in an overestimation of $\rho$. This indicates that the signature of GC events is similar to that of multiple CO events. We note that the estimated $\rho$ was close to the predicted rate when GC events would be counted as two CO events (+1 slope on Fig. 4).

LD-based methods infer an intermediate rate, lower than the total recombination rate but higher than the CO rate, in particular with a larger sample size (Fig. 9). They are, however, sensitive to the track length: when a longer track length of 2 kb was used, *pyRho* tended to behave like *rhoSMC* and overestimated $\rho$ (Supplementary Fig. 10).

Conclusions:

1) GC has a strong impact on the inference of the genome-average $\rho$.
2) The signal of GC is intermingled with that of recombination rate variation.
3) The *heRho* method can disentangle GC and CO events when the recombination rate is homogeneous but is biased when it is heterogeneous or when the population size is not constant in time.
4) In the presence of GC with relatively short track lengths, the LD-based method *pyrho* provides an estimate close to the CO rate. When the track length becomes longer, $\rho$ is overestimated.
5) *rhoSMC* overestimates $\rho$ in the presence of GC, independently of the track length.

## Testing for the occurrence of recombination

Lastly, we explored whether recombination rate inference methods could be used as tests for the occurrence of recombination; that is, we more specifically assessed their performance when the simulated recombination rate was zero. In the absence of recombination, the *pyrho* method was unbiased and consistently returned a very low $\rho$ value (Supplementary Fig. 11). *LDhat* returned a low rate when a small sample size was used but takes an unusually long time to converge for larger datasets (more than 48 h per simulation). When a single diploid was used, *heRho* failed to estimate parameters in all but one case and returns a CO rate of 0.5 and a GC rate of 1. When at least five diploids were used, the method returned a zero CO rate in all but one case and a non-zero GC rate.

SMC-based methods showed distinct behaviors: *MSMC*, which relies on the Baum–Welch algorithm, either inferred a low rate if a single diploid was used or consistently failed to converge (when five diploids were used). *iSMC* and its variant *rhoSMC* returned an unrealistically high rate in almost all replicates when a single diploid was used and failed to converge in one replicate when five diploids were used. This is because the sample shares a unique ancestor for all positions, preventing the fit of a distribution of divergence times when a single individual is used. We note that the key parameter here is the number of chromosomes rather than the actual number of individuals. The composite likelihood approach generates identical likelihood values whether two chromosomes come from the same or two distinct individuals since the Markov chain is reset and the likelihoods for each

chromosome are multiplied. When several chromosomes from one individual are used, even if the intra-chromosome recombination rate is zero, the inter-chromosomal rate would be equal to 0.5 and distinct chromosomes will have different ancestors, resulting in a better fit of the model.

Conclusions:

1) If the dataset consists of a single chromosome of a single individual and the simulated recombination rate is zero, all methods will provide erroneous estimates.
2) *pyRho* is the most reliable method to infer very low recombination rates for intermediate or large sample sizes.

## Discussion

### The LD-based method *pyrho* performs best over a large range of conditions, but SMC methods offer a promising alternative for (very) small sample sizes

While estimating the genome average population recombination rate $\rho = 4 \cdot Ne \cdot r$, a single parameter, may seem a simple endeavor in comparison to inferring a detailed recombination map, the task is challenging because *r* varies spatially and *Ne* temporally. (Not to mention the possibility that *r* may also vary temporally and *Ne* spatially, two possibilities that we have not investigated in this study.) LD-based methods (particularly the most recent ones such as *Pyrho*) that account for nonconstant demography proved to provide the most accurate estimations under all tested scenarios. However, their power is maximal at relatively large sample sizes. Conversely, SMC-based methods offer a powerful alternative when only one unphased diploid genome is available. Such methods properly account for the demography of the underlying population, which is jointly estimated with recombination rate parameters. Two factors appear important when inferring $\rho$ with SMC models: using a full maximum-likelihood estimation procedure instead of the Baum–Welch algorithm and properly accounting for the recombination rate heterogeneity along the sequence. SMC-based methods, however, rely on the assumption that a reasonable amount of meiotic recombination occurs and can return unrealistic estimates when this is not the case. This bias, however, is mitigated when more than one chromosome is analyzed, either from the same individuals or distinct ones.

### Sources of bias

SMC models cannot capture the recombination hotspot signal and only infer the background recombination rate. As recombination hotspots represent by nature very fine-scale variation, their signal may only be captured with large sample sizes. As of today, only LD-based methods are powerful enough to capture this signal.

Population structure and admixture events were reported to bias LD-based estimates of recombination rates (Samuk and Noor 2022); it will be important to assess how they affect the inference of $\rho$ with *heRho* and SMC-based methods in future studies.

We further showed that the presence of GC significantly biases the inference of $\rho$. LD-based methods essentially capture the CO rate and some of the GC events, although only when the mean GC track length is short. SMC-based methods show complex behavior in the presence of GC. When the recombination rate is homogeneous, *iSMC* behaves like LD methods and infers intermediate rates, although the estimation is impacted by the underlying demography, pointing at some interdependence between these parameters. *MSMC*'s demography inference is largely unaffected under 50% of GC events and is still accurate but with increased variance up to 90% GC events (Supplementary Fig. 12).

Whether this effect results from GC itself or the reduced amount of CO events would deserve further investigation.

The *rhoSMC* method, however, shows a strong bias and tends to infer two CO events for one GC event, leading to a significant overestimation of $\rho$ when the proportion of GC events is high. This result is particularly concerning as the relative rate of GC vs CO has been suggested to vary extensively between genomic regions and could be as high as 100× (Padhukasahasram and Rannala 2013).

### Disentangling CO and GC events

The *heRho* method is currently the only one that can infer both the CO and total recombination rates in the presence of GC. However, in its current implementation, it can only do so when the recombination landscape is homogeneous and the demography constant—two conditions unlikely to be met in a majority of cases. In the original publication (Setter *et al.* 2022), the authors assess the impact of rate heterogeneity by mixing two homogeneous datasets with distinct recombination rates and did not find a significant impact on the estimation. Here, we used a more realistic scenario where the recombination rate takes values from an exponential distribution or from an empirical distribution (the deCODE dataset) and found a significant impact of recombination rate heterogeneity on the inference of the average rate.

While nonconstant demography may be incorporated in the modelling, the signal of GC and variable recombination rate may be difficult to disentangle, as GC generates short homozygous segments—a typical signature of highly recombining regions. Incorporating GC into the SMC model framework offers an exciting perspective, if feasible. Notably, De Maio and Wilson (2017) implemented an SMC model with GC only, which they refer to as a "bacterial SMC". While the model was used for simulations only, the authors demonstrated that it could be combined with an approximate Bayesian computation framework to infer GC rates and track lengths. Future studies will have to show whether such parameters are still identifiable in the presence of both CO and GC events.

## Acknowledgments

The author would like to thank Gustavo Barroso, Bernhard Haubold, and Linda Odenthal-Hesse for discussions on this project, as well as Konrad Lohse, Derek Setter, and one anonymous reviewer for constructive comments on a previous version of this article. This project emerged from discussions with Gwenael Piganeau and Victor Loegler. The idea of assessing the impact of gene conversion on recombination inference was suggested by Richard Durbin, after a presentation by the author at the PopGroup meeting in 2022.

## Data availability

All the necessary code to reproduce the analyses and figures in this study, as well as intermediate result tables, are available at https://gitlab.gwdg.de/molsysevol/smc-benchmark. The version used in this manuscript is also available at FigShare, DOI: 10.6084/m9.figshare.24069270.

Supplemental material available at GENETICS online.

## Funding

JYD acknowledges funding from the Max Planck Society. This work was supported by a grant from the German Research Foundation (Deutsche Forschungsgemeinschaft) attributed to JYD, within the priority program (SPP) 1590 "probabilistic structures in evolution". The funders had no role in study design, data collection and analysis, decision to publish, or preparation of the article.

## Conflicts of interest

The author has declared that no competing interests exist.

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

*Editor: K. Lohse*