## [Peer Review File · Genetics]

On the estimation of genome-average recombination rates

Julien Dutheil

NOTE: The reviews and decision letters are unedited and appear as submitted by the reviewers.

In extremely rare instances and as determined by a Senior Editor or the EIC, portions of a review may be redacted. If a review is signed, the reviewer has agreed to no longer remain anonymous.

The review history appears in chronological order.

Review Timeline:

Submission Date:	2023-09-02
Editorial Decision:	2023-09-30
Resubmission Received:	2024-01-22
Editorial Decision:	2024-02-16
Revision Received:	2024-03-13
Accepted:	2024-03-20

September 30, 2023

GENETICS-2023-306468

On the estimation of genome-average recombination rates

Dear Dr. Dutheil:

Two experts in the field have reviewed your manuscript, and I have read it as well. I agree with the reviewers that investigating how well widely used popgen inference methods can infer recombination under biologically plausible settings (GC, variable demography and recombination) is an important contribution to the field.

While your manuscript is not currently acceptable for publication in GENETICS, we would welcome a substantially revised manuscript. Both reviewers (in particular reviewer 1) have comments and concerns to be addressed. You can read their reviews at the end of this email. Please make sure to address the following issues in a revised version:

- The central question about methods that co-estimate recombination and demographic history is which combinations of processes (heterogeneity in N_e , heterogeneity in ρ , GC) are identifiable, in particular, when inference is based on a single genome. The current sole focus of the MS on the average recombination rate does not really get at this wider question. Please include results for the power to infer demography (for iSMC and MSMC) and recombination heterogeneity (for iSMC) (see reviewer 1). It would be really helpful to better understand the difference in performance of iSMC and MSMC; i.e. does the bias in ρ estimates of MSMC simply reflect the fact that this method was developed to infer demography (and uses many parameters to model it), whereas a major motivation for the development of iSMC was inference of ρ (and so demography is modelled with fewer parameters)? Should researchers who are primarily interested in demographic history worry about the ρ biases of MSMC?
- the manuscript can be condensed: several of the main figures can be combined (see reviewer 1) and some sections can be shortened. E.g. quite a bit of text is spent on describing what happens when $\rho = 0$. Surely all methods will fail in this case, so why is this relevant?
- there are many organisms without recombination hotspots or substantial ρ heterogeneity, while GC and variable demography seem ubiquitous. Please include a scenario of variable demography and GC (but no heterogeneity in ρ). This would also give a better understanding of why methods are biased when both ρ and N_e vary.
- please make the simulation code used available. This will allow applying the same benchmarking to any future methods and to understand what exactly was done. For example, it was not clear to me how GC was simulated: msprime seems to only allow for exponentially distributed GC tracts, whereas the MS states that "a constant tract length of 300 bp" was specified.
- better motivate the choice of parameter space explored: for some parameters, values that are plausible for humans were chosen (e.g. μ , recombination) but not for all (N_e). While it is useful to investigate performance for a broad range of parameter space (in particular ρ/μ), it would be helpful to indicate (both in the text and figures) which parameter combinations are relevant for which kind of systems (e.g. humans, *Drosophila*).

We look forward to receiving your revised manuscript. Please let the editorial office know approximately how long you expect to need for revisions.

Upon resubmission, please include:

1. A clean version of your manuscript;
2. A marked version of your manuscript in which you highlight significant revisions carried out in response to the major points raised by the editor/reviewers (track changes is acceptable if preferred);
3. A detailed response to the editor's/reviewers' feedback and to the concerns listed above. Please reference line numbers in this response to aid the editor and reviewers.

Your paper will likely be sent back out for review.

Additionally, please ensure that your resubmission is formatted for GENETICS
<https://academic.oup.com/genetics/pages/general-instructions>

Follow this link to submit the revised manuscript: Link Not Available

Sincerely,

Konrad Lohse

Associate Editor
GENETICS

Approved by:
Nicholas Barton
Senior Editor
GENETICS

Reviewer #1 (Comments for the Authors (Required)):

In this manuscript, the author examines the ability of a number of methods to infer average recombination rates under a number of 'non-ideal' scenarios. They compare SMC-based methods (MSMC, iSMC), LD-based methods (LDHat, Pyrho), and heterozygosity-based methods (heRho). They illustrate the base performance of each under a model of constant demography and homogenous recombination rates, then consider the effect that recombination rate heterogeneity, complex demography, recombination hotspots, and gene conversion have on each method. They show that: SMC methods must model recombination rate heterogeneity to obtain an accurate prediction of the genome-wide average while spatially unaware LD-based methods still perform well in that respect. Complex demography biases estimates for demography-unaware methods like Ldhat and heRho. The effect of recombination hotspots is only captured by LD-based methods, though with hotspots, SMC-based methods still estimate the 'background' rate (outside hotspots). Only the heterozygosity-based method heRho performs well when gene conversion is present but then only under a constant demography. In conclusion, the author calls for the incorporation of gene conversion into an SMC framework.

I think this is a timely, informative, and generally well-written paper. Given their wide use, it is helpful for researchers to understand the limitations and caveats of applying these methods, and this manuscript investigates their performance under biologically realistic scenarios that have largely been ignored to date. I wholeheartedly agree that it would be exciting to incorporate gene conversion into SMC methods.

That said, I do have some reservations about the manuscript. Some sections are a bit verbose, the figures are rather repetitive, and there are a few key results missing which would make this paper more impactful. Below, I have listed both major and minor concerns as well as some suggestions for improving the text and figures.

Major:

There are a lot of choices made about simulation parameters and procedures for the various scenarios, but they mostly lack a connection to real biology. Are the rate parameters similar to any particular species? What about the way variable recombination landscapes and hotspots are generated? Why did you choose those demographic models? Even some motivation from a theoretical perspective would help (i.e. have you picked params/scenarios that particularly push the boundaries of the methods?).

I realize that the point of the paper is about how well different SMC methods do at estimating the average recombination rate, but it begs the question whether these methods are still able to infer an accurate demographic history under the different recombination scenarios (heterogeneity, hotspots, and gene conversion). It is also highly relevant information for researchers to interpret the results they get using these methods. I'd like to see a figure or two showing the performance of the SMC-based methods when inferring the demography in these cases.

Again, although the paper is about average rate estimates, I would also like to see how rhoSMC performs in resolving the underlying recombination rate map under the hotspots and gene conversion scenarios.

The "Inference under an ideal scenario" is a bit verbose. I wonder if the paragraphs about getting MSMC to work could be shortened or moved to an appendix/supplement, and I wonder whether too much attention is paid to the rho=0 case.

P 7 | 26-28: I'm not exactly sure what 'properly captures the heterogeneity' means here. It doesn't seem to perform better than Ldhat or Pyrho. The only difference between these seems to be that rhoSMC more or less estimates the CO rate + 2*(GC rate), while LDHat and Pyrho actually capture something reasonably close to the CO rate. I think 'properly captures the heterogeneity' would mean that it does a good job of capturing the fine-scale variation along the chromosome, which I don't think is shown in this paper.

The inference under a non-constant demography uses the variable recombination rate scenario, and this makes it difficult to

disentangle their effects. It would be nice to see results for the homogenous recombination + demography scenarios to clarify this (could be plotted on the same figure but in different colour).

Minor:

I had a look at the msprime docs, and for the GC scenario, I am a bit suspicious whether a constant tract length was simulated or rather this was the mean of a tract length distribution. If it is true that a constant tract length was used, what would happen if you did simulate a tract length distribution? Would we still see the same trends in figure 5 (particularly for rhoSMC) when tract length is a random variable?

The figures become really repetitive, and I think some of them could be combined together. E.g. combine Figure 1 and Figure 2 using colours to distinguish homogenous and heterogeneous. Same could be done for the separate panels of Figure 5

I also wonder whether the paper would be more concise and clear if MSMC results were largely removed. That is, address the optimization-based fault with MSMC in the "ideal scenario" section, then for the rest of the scenarios, only show the iSMC results.

abstract line 4 and other places: classical → classic (classical is for the type of music)

In the introduction, it would be good to distinguish the SMC approximation from methods that use it for inference, citing MSMC as well as iSMC in the 5th paragraph.

There should be a brief description of the gene conversion simulations in the methods section.

p1 l40: upon → about

p2 l 25: Does this variable landscapes model generate auto-correlation along the genome?

P2 l 33: chromosome 1 of humans, right?

P2 l 38: I had to read this several times to understand the model, but I think it's really simple. Maybe reword this a bit? As I understand it, you make a variable recombination landscape, normalize to a focal recombination rate, then go back and sprinkle small regions with a much higher rho... What is the expected average rho across the chromosome after the hotspots are added? Making this clearer will help the reader digest the results on page 6 much better.

P2 l 52: How does the decreasing pop size scenario affect genetic diversity relative to the constant size scenario?

P2 l 53: "asymmetric" → "opposite" or perhaps "converse"?

P 2 l 63: what do you mean by 'true' population recombination rate?

P5 l 1-2: This suggests that the variable landscape model DOES have an auto-correlation that matches the assumptions of the rhoSMC model. Is that true? See comment above about p2 l 25

P5 l 15-16: Might be clearer to say in the main text that 6/10 were off the scale.

P5 L 52: make it clear that this is for SMC still

p6 l 47: I think this should cross-ref to Fig 4 of the main text. It might help to explicitly define the different rho values here in the text ('simulated' vs 'inferred', vs 'true'), and clarifying the methods section will make this easy (see comment above).

Figure 5- the axis is hard to understand. Maybe it would be easier to plot these with their true values: $\rho = CO + GC = 0.06$ as the horizontal line, the CO rate alone (the current -1 slope), and the $2*\rho - CO$ line (+1 slope). And use different colours for ease. It should be clear that the (-1 slope) line is the correct CO rate. Maybe also useful to clarify column names to say "heRho CO" and "heRho CO + GC", because it took me a while to realize that the 'heRho' column SHOULD follow the -1 slope and the 'heRho total' column should track the centre line.

Supplementary Notebook

For the figure in the "Check demography" section. What do the rows labelled 1 to 5 represent? And What are the different lines within each plot?

Reviewer #2 (Comments for the Authors (Required)):

This manuscript is about benchmarking methods in inferring recombination rates, under different scenarios, including demography, rate heterogeneity and confounding from gene conversion. This work is, without doubt, an important contribution to the community as there is a lack of comprehensive look into all these methods under all these realistic circumstances. However, I would still like to give some suggestions to improve the reading experience from potential readers and the discussion depth about explanation method performances. The 2 biggest advice I had are: 1) make sure the benchmark results are well presented in a more concise way (tables of MSE/correlation) so that it could be read easily 2) explain the reason of certain methods not working in certain scenarios from their algorithm design, rather than description of the behavior.

1. A general comment for all benchmark results/figures in this manuscript is that it would be nicer to have some numbers/values for benchmarking results, such as the MSE/correlation between ground truth and inference results. I would also recommend having a table summarizing each figure. Otherwise seeing from the plots themselves is a bit tricky;

2. For SMC based methods, does it help if you first estimate a recombination map and then use this inferred map to the model and iterate a few times? It is generally difficult to jump from a uniform prior to heterogeneity, but might be more possible with a few updates. I remember there is a figure S6 in ARGweaver paper showing that a hotspot can be detected, but to a lesser extent. I wonder if this could be helped by updating the map and iterating. I understand that it is too much to test everything out, but a small demonstration or conceptual discussion would be ideal;

3. For the non-constant demography section, I am curious why out of SMC-based methods only rhoSMC can handle non-constant demography (figure 3). Technically MSMC and iSMC can be defined under a model (that's how PSMC is used), but they cannot seem to handle the demography knowledge. Do they not take into the demography or simply not work even with demography? I think it is worth mentioning the reason why they underperform, rather than merely describing their behavior;

4. For the recombination inference with gene conversion section, I think it deserves more discussion on the reason behind their performance, instead of simply describing their behavior. For example, mistaking 1 GC for 2 CO events makes a lot of sense for rhoSMC, but why isn't this the case for iSMC and MSMC, which are both SMC-based methods? It is also interesting to see that LD-based methods seem to have inference the CO rate decently accurately, some explanations on why CO is not contributing to LD signal for recombination detection might be interesting and worth adding. (Personally I think it has to do with the average GC track length, if that is longer, it might screw things up for LD-based methods, the reason why they didn't "see" them might be because the tract length is smaller than the typical distance between informative SNP pairs);

5. For many methods which can not really handle demography or rate heterogeneity, it is worth discussing whether there is fundamental difficulty in adding that into the corresponding framework, or it could be done easily but not added yet. This discussion would be hugely beneficial for future methods development.

Overall this works fulfills an important task of benchmarking recombination inference with quite some realistic considerations. I believe that with further modifications this work would be well appreciated and referenced. Thanks to the author for collecting and contributing the knowledge!

Associate Editor Comments:

Minor comments:

- a recent simulation study (Samuk and Noor 2022 G3) investigates the impact gene flow between diverging populations on the performance of some of these methods. It seems important to at least mention this other simulation study.

- page 7, l. 60 in case -> when

Dear editor and reviewers,

I am very grateful for the comments and suggestions on my manuscript. In this revised version, I have tried to address them and I believe the manuscript has gained in clarity as a result. As a summary, this revised version include:

- New simulations under non-constant demography and a flat recombination landscape
- New simulations with a GC track length of 2 kb on average instead of 300 bp

The manuscript content was restructured to:

- Suppress the “Inference under a non-constant demography” section. Instead, the impact of demography is now discussed both in the flat and variable recombination landscape sections.
- Discuss the $r = 0$ case in a new dedicated section (the “ideal scenario” section was then shortened and simplified as a result).

I answer below in more details each point raised.

- The central question about methods that co-estimate recombination and demographic history is which combinations of processes (heterogeneity in N_e , heterogeneity in ρ , GC) are identifiable, in particular, when inference is based on a single genome. The current sole focus of the MS on the average recombination rate does not really get at this wider question. Please include results for the power to infer demography (for iSMC and MSMC) and recombination heterogeneity (for iSMC) (see reviewer 1).

- ⇒ The revised version includes new (supplementary) figures showing the inferred demographics, both under a flat and variable recombination landscapes. While the scenarios tested in this work are most likely too simplistic to permit generalization, the conclusions reached are consistent with those of previous works. Globally, demography inference is robust to recombination rate heterogeneity (Li and Durbin 2011; Schiffels and Durbin 2014). One new result, however, is that demography inference no longer works when the recombination rate is high and homogeneous, while it is robust to the presence of hotspots. I have emphasized this conclusion in the revised version.

In this study, I focused on the genome-average recombination rate, and did not characterize the inferred recombination map itself. The SMC-based methods and heRho do not infer a map, but a single parameter for the genome. rhoSMC can be used to infer a map, when used in combination with a posterior decoding procedure. I have now compared the rhoSMC and pyRho inferred maps to the simulated one and assessed their correlation. See the corresponding answer to reviewer 1 for more details.

It would be really helpful to better understand the difference in performance of iSMC and MSMC; i.e. does the bias in rho estimates of MSMC simply reflect the fact that this method was developed to infer demography (and uses many parameters to model it), whereas a major motivation for the development of iSMC was inference of rho (and so demography is modelled with fewer parameters)? Should researchers who are primarily interested in demographic history worry about the rho biases of MSMC?

⇒ To address this point, I added a new set of results where I ran MSMC on the “ideal” scenario with 30 intervals, but a single population size (that is, enforcing a flat demography). This did not remove the bias, only did it improve the estimation when $r = 0$. So the parametrization of the demographic model is not the issue here. Yet, when demography inference is the primary goal, the proper estimation of ρ is not required. In the revised version of the manuscript, I report more details about the inferred demography and show that it is generally well recovered by MSMC, showing that it is robust to the actual inferred recombination rate (consistent with what was reported in the original PSMC and MSMC articles).

- the manuscript can be condensed: several of the main figures can be combined (see reviewer 1) and some sections can be shortened. E.g. quite a bit of text is spent on describing what happens when $\rho = 0$. Surely all methods will fail in this case, so why is this relevant?

⇒ I have combined the figures as suggested, including all demographic scenarios. To increase visibility, I used color and summarized all 10 replicates with averages and 95% confidence intervals. The $\rho = 0$ case is interesting as for some non-model organisms, it might not even be known whether they undergo sexual reproduction at all. Reporting a non-null recombination rate might be used as evidence for the occurrence of sexual reproduction. I show here that some caution should be taken in doing so, in particular when using small sample sizes, as SMC-based methods may return high recombination rate estimates in some circumstances. In the revised version, I moved this discussion in a dedicated section and made the focus of the study more explicit in the introduction, including testing for the occurrence of recombination.

- there are many organisms without recombination hotspots or substantial ρ heterogeneity, while GC and variable demography seem ubiquitous. Please include a scenario of variable demography and GC (but no heterogeneity in ρ). This would also give a better understanding of why methods are biased when both ρ and N_e vary.

⇒ In the revised version, I have now included results for simulations under the three demographic scenarios for both the homogeneous and non-homogeneous recombination landscapes, with and without gene conversion. These results are summarized in updated figures which display all three scenarios together.

- please make the simulation code used available. This will allow applying the same benchmarking to any future methods and to understand what exactly was done. For example, it was not clear to me how GC was simulated: msprime seems to only allow for exponentially distributed GC tracts, whereas the MS states that "a constant tract length of 300 bp" was specified.

⇒ I apologize for the error when reporting GC simulations. A new dedicated section has been added in the Materials and Methods of the revised version.

All scripts and intermediate results are provided in the GitLab repository mentioned in the Data Availability section.

- better motivate the choice of parameter space explored: for some parameters, values that are plausible for humans were chosen (e.g. μ , recombination) but not for all (N_e). While it is useful to investigate performance for a broad range of parameter space (in particular ρ/μ), it would be helpful to indicate (both in the text and figures) which parameter combinations are relevant for which kind of systems (e.g. humans, *Drosophila*).

⇒ A justification of the choice of parameter has been added in the revised version, see corresponding answer to reviewer 1.

Reviewer #1

I think this is a timely, informative, and generally well-written paper. Given their wide use, it is helpful for researchers to understand the limitations and caveats of applying these methods, and this manuscript investigates their performance under biologically realistic scenarios that have largely been ignored to date. I wholeheartedly agree that it would be exciting to incorporate gene conversion into SMC methods.

That said, I do have some reservations about the manuscript. Some sections are a bit verbose, the figures are rather repetitive, and there are a few key results missing which would make this paper more impactful. Below, I have listed both major and minor concerns as well as some suggestions for improving the text and figures.

⇒ Thank you for your constructive comments and positive assessment of this work.

Major:

There are a lot of choices made about simulation parameters and procedures for the various scenarios, but they mostly lack a connection to real biology. Are the rate parameters similar to any particular species? What about the way variable recombination landscapes and hotspots are generated? Why did you choose those demographic models? Even some motivation from a theoretical perspective would help (i.e. have you picked params/scenarios that particularly push the boundaries of the methods?).

⇒ The first paragraph of the result was extended to include a justification of the choice of parameter values. It now reads as:
“The simulated sequences always consist of a unique chromosome of size 10 Mb and ten replicates. The mutation rate was set to that of humans, $1.25e-8$ bp⁻¹ and kept constant in all analyses. The combination of the mutation rate parameter and chromosome length ensures that there is enough diversity to fit models while keeping the computational time feasible over multiple replicates. The 10 Mb chromosome length also permits assessing the methods' accuracy on small genomes such as the ones of several micro-

organisms. The number of individuals used for inference, the demographic scenario, recombination landscape and inference method vary in each experiment. The range of average recombination rates (ρ in $[0, 0.03]$) and landscapes (flat, heterogeneous, deCODE map, with hotspots) covers a broad range of biological conditions. Three simple demographic scenarios are tested: constant population size, population decline and growth. The focus of this work is not to assess the ability of the methods to infer demography but rather to evaluate the robustness of recombination inference to deviations, be them simple, from a constant population size.”

I realize that the point of the paper is about how well different SMC methods do at estimating the average recombination rate, but it begs the question whether these methods are still able to infer an accurate demographic history under the different recombination scenarios (heterogeneity, hotspots, and gene conversion). It is also highly relevant information for researchers to interpret the results they get using these methods. I'd like to see a figure or two showing the performance of the SMC-based methods when inferring the demography in these cases.

- ⇒ Supplementary Figure 1 has been extended to show the inferred demography under the three scenarios. In addition, two new figures (Supplementary figures 4 and 12) have been added, with a similar setting as Supplementary Figure 1, but under the variable recombination rate scenario and in the presence of GC, respectively. These results are now interpreted in the discussion. Globally, they confirm that the demography inference is robust to the actual recombination landscape, excepted when the recombination rate is very high, or when the proportion of GC events is elevated (>50% of recombination events).

Again, although the paper is about average rate estimates, I would also like to see how rhoSMC performs in resolving the underlying recombination rate map under the hotspots and gene conversion scenarios.

- ⇒ As discussed in the manuscript and in the original publication, rhoSMC is blind to hotspots. I compared the inferred maps at different scales to the simulated map, and plotted the results together with those of Pyrho, using 5 diploids:

(Median correlation over 10 replicates, the error bars show the min and max measured correlations.) With this sample size, Pyrho is also missing some of the hotspots.

As for the effect of gene conversion, both methods are equally affected:

While I find these results interesting, I believe they are falling a bit outside the scope of the present paper, which focuses on the genome average. I feel like the impact of GC on the inference of recombination maps could be the topic of a study in itself, and that more analyses on the matter could/should be conducted. I would, therefore, prefer not to include these results (these figures and corresponding scripts are nonetheless included in the GitLab repository), but could do so if the editor and reviewers deem this valuable / necessary.

The "Inference under an ideal scenario" is a bit verbose. I wonder if the paragraphs about getting MSMC to work could be shortened or moved to an appendix/supplement, and I wonder whether too much attention is paid to the rho=0 case.

⇒ This section has been edited. The discussion about the particular case of rho = 0 has been simplified and is now moved to a dedicated new section.

P 7 | 26-28: I'm not exactly sure what 'properly captures the heterogeneity' means here. It doesn't seem to perform better than Ldhat or Pyrho. The only difference between these seems to be that rhoSMC more or less estimates the CO rate + 2(GC rate), while LDHAT and Pyrho actually capture something reasonably close to the CO rate. I thought 'properly captures the heterogeneity' would mean that it does a good job of capturing the fine-scale variation along the chromosome, which I don't think is shown in this paper.*

- ⇒ This conclusion inappropriately referred to the case where $GC = 0$. It was now updated to:
“rhoSMC overestimates the recombination rate in the presence of gene conversion, independently of the track length.”

The inference under a non-constant demography uses the variable recombination rate scenario, and this makes it difficult to disentangle their effects. It would be nice to see results for the homogenous recombination + demography scenarios to clarify this (could be plotted on the same figure but in different colour).

- ⇒ New analyses with increasing and decreasing population sizes for the flat recombination landscape have been included in the revised version of the manuscript. The new figures 1 and 2 are directly comparable, with three identical demographic scenarios used in combination with two recombination landscapes, homogeneous and variable.

Minor:

I had a look at the msprime docs, and for the GC scenario, I am a bit suspicious whether a constant tract length was simulated or rather this was the mean of a tract length distribution. If it is true that a constant tract length was used, what would happen if you did simulate a tract length distribution? Would we still see the same trends in figure 5 (particularly for rhoSMC) when tract length is a random variable?

- ⇒ This was indeed a mistake in the text, a variable track length was used. The msprime algorithm was used without any further modification. In the revised version, the corresponding section was amended and some additional data with a track length of 2 kb were analyzed for comparison.

The figures become really repetitive, and I think some of them could be combined together. E.g. combine Figure 1 and Figure 2 using colours to distinguish homogenous and heterogeneous. Same could be done for the separate panels of Figure 5

- ⇒ Thank you for this suggestion. Figures have been combined as suggested. Furthermore, I now used mean \pm 95% CI instead of plotting the raw data, which hopefully further improved their clarity.

I also wonder whether the paper would be more concise and clear if MSMC results were largely removed. That is, address the optimization-based fault with MSMC in the "ideal scenario" section, then for the rest of the scenarios, only show the iSMC results.

- ⇒ Thank you for this suggestion. MSMC results and their discussion have been removed from the figures concerning the estimation of rho, excepted from the first one, as suggested. As it is the only method whose original purpose is demography inference, MSMC is still discussed in this context.

abstract line 4 and other places: classical → classic (classical is for the type of music)

⇒ Corrected!

In the introduction, it would be good to distinguish the SMC approximation from methods that use it for inference, citing MSMC as well as iSMC in the 5th paragraph.

⇒ This section of the introduction was modified as suggested. It now reads as: “In parallel to the development of methods to infer recombination rates, models based on the sequentially Markov coalescent (SMC) unleashed the power of full genome data for demography inference (Spence et al. 2018). The SMC is an approximation of the sequential coalescent with recombination (Wiuf and Hein 1999). While it makes simplifying assumptions about the recombination process, SMC-based inference relies on the recombination rate, which can be jointly estimated with demographic parameters in inference methods like PSMC (Li and Durbin 2011) and MSMC2 (Schiffels and Wang 2020). This property has been further used in the iSMC method, which accounts for the heterogeneity of the recombination rate assuming a prior distribution and an auto-correlation process along the genome (Barroso et al. 2019).”

There should be a brief description of the gene conversion simulations in the methods section.

⇒ A section reporting on the gene conversion simulations was indeed missing and has now been added.

p1 l40: upon → about

⇒ Corrected.

p2 l 25: Does this variable landscapes model generate auto-correlation along the genome?

⇒ If I am correct, this model is equivalent to a continuous-state, discrete-time Markov process with a one-parameter transition rate matrix (Jukes Cantor-like, with $r = 1/10,000$).

P2 l 33: chromosome 1 of humans, right?

⇒ Yes, this is now explicitly stated.

P2 l 38: I had to read this several times to understand the model, but I think it's really simple. Maybe reword this a bit? As I understand it, you make a variable recombination landscape, normalize to a focal recombination rate, then go back and sprinkle small regions with a much higher rho... What is the expected average rho across the chromosome after the hotspots are added? Making this clearer will help the reader digest the results on page 6 much better.

- ⇒ Thank you for this suggestion. I have now rewritten this section to make it clearer. It now read as:
 “Hotspots were simulated using a model similar to that of the variable recombination landscape. Regions of uniform rates were generated, with a rate sampled from an exponential distribution of mean 1.0 and a length sampled from a geometric distribution with a mean of 50 kb. As in the variable recombination model, the “background” rate of these regions was then multiplied by the specified average recombination rate.
 Between each region, hotspots with a recombination rate of $70 \times 1.25e-8 \text{ bp}^{-1}$ were inserted, with a breadth sampled from a distribution uniform between 0.5 and 2 kb. In this model, the genome average recombination rate is a weighted average of the background rate, ρ_{bg} , and the recombination rate in the hotspot regions, $\rho_{hs} = 70 \times 1.25e-8 \text{ bp}^{-1}$:
- $$\rho = (1 - \lambda) \cdot \rho_{bg} + \lambda \cdot (\rho_{hs})$$
- where lambda is the proportion of the simulated sequence located within hotspots.”

P2 | 52: How does the decreasing pop size scenario affect genetic diversity relative to the constant size scenario?

P2 | 53: "asymmetric" → "opposite" or perhaps "converse"?

- ⇒ Given the extended discussion regarding demography in the revised version, this section has been rewritten and now include estimates of the resulting genetic diversity. It now reads as:
 “We considered two scenarios departing from a constant population size. In the first scenario (referred to as “population decline” in the following), an ancestral population size of $N_e = 100,000$ exponentially decreases to $N_e = 1,000$ at present time, starting 1,000 generations before present. This population decline had little impact on the resulting genetic diversity ($\theta = 0.004586$ vs. $\theta = 0.004940$, when a flat recombination landscape is used). In a second scenario (referred to as “population growth” in the following), we considered a scenario where an ancestral population of size 50,000 increased exponentially to a size of 200,000. We note that the genetic diversity in this growth scenario is roughly half that of the constant or decreasing population size scenarios ($\theta = 0.002600$), owing to the starting population size being 50,000 instead of 100,000 in the other two scenarios.”

P 2 | 63: what do you mean by 'true' population recombination rate?

- ⇒ This referred to the simulated recombination rate. The manuscript has been edited to replace “true rate” by “simulated rate”.

P5 | 1-2: This suggests that the variable landscape model DOES have an auto-correlation that matches the assumptions of the rhoSMC model. Is that true? See comment above about p2 | 25

- ⇒ This is partially true. The rhoSMC model is a discrete version of the one used for simulations. This paragraph was rephrased with the hope to make it clearer:

“The auto-correlation model used to generate the recombination map used in the simulations is very similar to the recombination rate variation model of *rhoSMC*, the latter using discrete rate categories instead of continuous recombination rates. To assess the performance of *rhoSMC* when the recombination landscape is not generated by this simple auto-correlation pattern, the analysis was repeated on a recombination landscape taken from the deCODE human recombination map”

P5 l 15-16: Might be clearer to say in the main text that 6/10 were off the scale.

⇒ Corrected as suggested.

P5 l 52: make it clear that this is for SMC still

⇒ Corrected as suggested.

p6 l 47: I think this should cross-ref to Fig 4 of the main text. It might help to explicitly define the different rho values here in the text ('simulated' vs 'inferred', vs 'true'), and clarifying the methods section will make this easy (see comment above).

⇒ There was an error in the previous version of the manuscript: Figure 4 (not Supplementary Figure 4) should have been referenced here. The legend of Figure 4 recalls the definition of the genome average and background rates, which are now properly defined in the method section. Furthermore, the text now consistently uses the terms “simulated rate” instead of “true rate”.

Figure 5- the axis is hard to understand. Maybe it would be easier to plot these with their true values: $\rho = CO + GC = 0.06$ as the horizontal line, the CO rate alone (the current -1 slope), and the $2\rho - CO$ line (+1 slope). And use different colours for ease. It should be clear that the (-1 slope) line is the correct CO rate. Maybe also useful to clarify column names to say "heRho CO" and "heRho CO + GC", because it took me a while to realize that the 'heRho' column SHOULD follow the -1 slope and the 'heRho total' column should track the centre line.*

⇒ Thank you for these suggestions. “heRho” was replaced by “heRho (CO)” and “heRho (total)” by “heRho (CO+GC)” in all figures. Unfortunately, plotting everything related to 0.06 is no longer possible in the updated figure (now Figure 4), as it combines results for the three demographic scenarios (which result in different rhos for the same r , because of distinct N_e). Plotting absolute values make the comparisons between scenarios difficult as the figure becomes messy. I hope the new figure design will help on their clarity. In addition, I have amended the figure legend, which now reads as:
« Inference of the genome-wide recombination rate in the presence of gene conversion (GC). Simulations under a recombination rate of 1.5 cM/bp with various proportions of GC events (x-axis). The y-axis reports the ratio of the inferred recombination rate to the simulated one as percentages. Facet rows display various scenarios, with a sample size of 5 diploids in all cases. Results for sample sizes of 1 are shown in Supplementary Figure 7 and 50 in Supplementary Figure 8. Horizontal dash lines correspond to the simulated relative recombination rate (100%), and the -1 slope dashed line shows the

CO rate as a proportion of the total recombination rate. The +1 slope dashed lines show the theoretical total recombination rate when GC events count as two recombination events. Methods falling on the $y = -x$ line correctly estimate the CO rate, while methods falling on the $y = 0$ horizontal line correctly estimate the total recombination rate. »

Supplementary Notebook

For the figure in the "Check demography" section. What do the rows labelled 1 to 5 represent? And What are the different lines within each plot?

- ⇒ These represent the number of diploids used when running MSMC. This is now specified.

Reviewer #2 (Comments for the Authors (Required)):

This manuscript is about benchmarking methods in inferring recombination rates, under different scenarios, including demography, rate heterogeneity and confounding from gene conversion. This work is, without doubt, an important contribution to the community as there is a lack of comprehensive look into all these methods under all these realistic circumstances. However, I would still like to give some suggestions to improve the reading experience from potential readers and the discussion depth about explanation method performances. The 2 biggest advice I had are: 1) make sure the benchmark results are well presented in a more concise way (tables of MSE/correlation) so that it could be read easily 2) explain the reason of certain methods not working in certain scenarios from their algorithm design, rather than description of the behavior.

- ⇒ Thank you for these helpful suggestions, that I have taken into account in the revised version (see details below).

1. A general comment for all benchmark results/figures in this manuscript is that it would be nicer to have some numbers/values for benchmarking results, such as the MSE/correlation between ground truth and inference results. I would also recommend having a table summarizing each figure. Otherwise seeing from the plots themselves is a bit tricky;

- ⇒ In the revised manuscript, all figures have been updated with the goal to improve their clarity. Averages and error bars are displayed instead of raw data, and several figures are combined to ease their comparison. The case $\rho = 0$ is now also treated separately for simplification. As the manuscript primarily focus on possible estimation biases, correlation coefficients or MSE may not allow to properly assess the underlying effect: in many situations, we observe a strong estimation bias, but the correlation with the simulated rate might still be very high (the slope is just not equal to 1 and/or the relation is not linear).

2. For SMC based methods, does it help if you first estimate a recombination map and then use this inferred map to the model and iterate a few times? It is generally difficult to jump from a uniform prior to heterogeneity, but might be more possible with a few updates. I remember there is a figure S6 in ARGweaver paper showing that a hotspot can be detected, but to a lesser extent. I wonder if this could be helped by updating the map and iterating. I understand that it is too much to test everything out, but a small demonstration or conceptual discussion would be ideal;

- ⇒ While running a SMC-model on a given recombination map is conceptually simple, to my knowledge, it is not implemented in any software. A possible reason is that it would break down some of the implemented optimizations for the likelihood calculation (basically, the transition matrix between TMRCA categories changes along the genome, so that transition probabilities can no longer be pre-computed to save time). Such an approach would be relevant for demography inference. However, demography parameters seem to be quite robust to the recombination model, a finding that I confirmed in these simulations and now discuss in more details in the revised version.

3. For the non-constant demography section, I am curious why out of SMC-based methods only rhoSMC can handle non-constant demography (figure 3). Technically MSMC and iSMC can be defined under a model (that's how PSMC is used), but they cannot seem to handle the demography knowledge. Do they not take into the demography or simply not work even with demography? I think it is worth mentioning the reason why they underperform, rather than merely describing their behavior;

- ⇒ The underperformance of iSMC and MSMC is not due to the demography, but to the recombination heterogeneity. This is now apparent in the new Figures 1 & 2: when the landscape is homogeneous, iSMC infers the correct rate under all three demographic scenarios (Figure 1), while MSMC still suffers from the Baum-Welch estimation bias. Under a heterogeneous recombination landscape, iSMC under-estimates the recombination rate, but the underestimation is identical in all three demographic scenarios (Figure 2). These conclusions are stated explicitly in the revised version.

4. For the recombination inference with gene conversion section, I think it deserves more discussion on the reason behind their performance, instead of simply describing their behavior. For example, mistaking 1 GC for 2 CO events makes a lot of sense for rhoSMC, but why isn't this the case for iSMC and MSMC, which are both SMC-based methods? It is also interesting to see that LD-based methods seem to have inference the CO rate decently accurately, some explanations on why CO is not contributing to LD signal for recombination detection might be interesting and worth adding. (Personally I think it has to do with the average GC track length, if that is longer, it might screw things up for LD-based methods, the reason why they didn't "see" them might be because the tract length is smaller than the typical distance between informative SNP pairs);

- ⇒ Thank you for these comments. In the revised version, I have extended the discussion on the effect of biased gene conversion. I have conducted extra sets of simulations with a GC track of 2 kb. As the reviewer predicted, under

these conditions, pyRho started to “see” GC events and overestimated the recombination rate. SMC methods, however, were relatively unaffected by the track length.

Why iSMC does not overestimate rho under a flat landscape and the presence of GC is unclear to me. In the new updated figure, there seems to be an interaction between the presence of GC and demography, which could indicate a complex inter-dependence between these parameters. The demography itself, however, is properly inferred if gene conversion remains below 50% of recombination events (new Supp Figure 12).

5. For many methods which can not really handle demography or rate heterogeneity, it is worth discussing whether there is fundamental difficulty in adding that into the corresponding framework, or it could be done easily but not added yet. This discussion would be hugely beneficial for future methods development.

- ⇒ The last part of the discussion was extended in order to develop this aspect further. It now reads as:
“While non-constant demography may be incorporated in the modelling, the signal of GC and variable recombination rate may be difficult to disentangle, as GC generates short homozygous segments -- a typical signature of highly recombining regions. Incorporating GC into the SMC model framework offers an exciting perspective, if feasible. Notably, De Maio and Wilson (2017) implemented an SMC model with GC only, which they refer to as a “bacterial SMC”. While the model was used for simulations only, the authors demonstrated that it could be combined with an approximate Bayesian computation framework to infer gene conversion rates and track lengths. Future studies will have to show whether such parameters are still identifiable in the presence of both CO and GC events.”

Overall this work fulfills an important task of benchmarking recombination inference with quite some realistic considerations. I believe that with further modifications this work would be well appreciated and referenced. Thanks to the author for collecting and contributing the knowledge!

- ⇒ Thank you for the constructive comments and suggestions!

Associate Editor Comments:

Minor comments:

- a recent simulation study (Samuk and Noor 2022 G3) investigates the impact gene flow between diverging populations on the performance of some of these methods. It seems important to at least mention this other simulation study.

- ⇒ Thank you for suggesting this very relevant reference. The discussion now includes a new paragraph:
“Population structure and admixture events were reported to bias LD-based estimates of recombination rates (Samuk and Noor, 2022). As SMC-based methods and *heRho* permit the inference of recombination rates from very small samples, it is possible to pre-filter the data to obtain homogeneous sub-samples that show no or little population structure. Nevertheless, as past

admixture events might still represent significant model violations that may impact the inference of recombination rates, it will be important to assess how they affect other methods in future studies.”

- page 7, l. 60 in case -> when

⇒ Corrected.

References :

De Maio N., and D. J. Wilson, 2017 The Bacterial Sequential Markov Coalescent. *Genetics* 206: 333–343. <https://doi.org/10.1534/genetics.116.198796>

Li H., and R. Durbin, 2011 Inference of human population history from individual whole-genome sequences. *Nature* 475: 493–496. <https://doi.org/10.1038/nature10231>

Schiffels S., and R. Durbin, 2014 Inferring human population size and separation history from multiple genome sequences. *Nat Genet* 46: 919–925. <https://doi.org/10.1038/ng.3015>

February 16, 2024

RE: GENETICS-2024-306814

Dear Dr. Dutheil:

I am pleased to accept your manuscript entitled "On the estimation of genome-average recombination rates" for publication in GENETICS, pending minor revision.

Please submit your revision along with a brief description of how you modified the manuscript in response to the reviewers' concerns and suggestions (which can be viewed at the bottom of this email. Most important are: i) a careful revision of the introduction to better motivate the study, ii) a general fine-combing to condense the writing and fix english grammar issues (see my comments below), iii) addressing the concern of reviewer 2 regarding the prevalence of GC. The fact that demographic inference appears very sensitive to plausible rates of GC is an important result that should be highlighted, vi) checking Figure 3 matches the caption v) revising subheadings.

I expect you should be able to submit a revised manuscript within 30 days. A suitably revised manuscript will be acceptable for publication; I don't expect to send it out for review.

When revising the ms., please make an effort to shorten it, because that almost always improves a manuscript. We urge authors to heed the advice of Strunk and White: "omit needless words"¹. Follow this link to submit the revised manuscript: Link Not Available

Thank you for submitting this story to Genetics.

Sincerely,

Konrad Lohse
Associate Editor
GENETICS

Approved by:
Nicholas Barton
Senior Editor
GENETICS

Reviewer comments:

Reviewer #1 (Comments for the Authors (Required)):

I really appreciate the author's thoughtful and thorough effort both to address the reviewers' comments/concerns and to improve the text and figures. It was well worth it! I think the manuscript is in great condition now.

Thanks a ton for the additional results you provided me in your response. That was very considerate and definitely valuable. It is indeed rather tangential from the focus of the manuscript, so I agree with your decision to exclude them from the revised version.

I have no further concerns major or minor. (Note, however, that I did not closely proofread the revised text.)

Best regards,
Derek Setter

Reviewer #2 (Comments for the Authors (Required)):

I am glad to see that the author have addressed all of my previous comments, and there are only two few minor comments I had.

In response to comment 4, the author said:

"The demography itself, however, is properly inferred if gene conversion remains below 50% of recombination events"

However, as far as I know, gene conversion is believed to be more prevalent than crossovers. Mechanistically, both crossovers and gene conversion are induced firstly from double strand break, then from the break it is more likely to initiate a conversion than a crossover. So it is slightly concerning to see that the demography inference only works for low-conversion rates. A more meaningful benchmark with more realistic parameters might be needed here. However, please do correct me if I interpret the scenario wrong.

The other thing is that when discussing the effects of gene conversion, it remains unclear to me that how gene conversion tract length comes into play. Two simulation with the same conversion rate but different tract length will definitely show different behaviors. I would like to see some clarifications on this, and maybe the gene conversion rate has to be normalized by tract length to be comparable when tract length are different? What is the most reasonable way of combining conversion rate and tract length here to describe its net effects?

Associate Editor comments:

-
- The intro should be revised carefully to motivate the study a bit better:
 - It seems important to say that the mean recombination rate varies widely across taxa to motivate the question of how best to infer it. It may be useful to include a sentence about recombination rate variation across taxa when citing Stapely et al 2017.
 - l. 13 "... involving various degrees of laboratory work (Peñalba and Wolf 2020)." Please distinguish more clearly between direct and indirect methods for inferring recombination rates here (and highlight the different timescales involved). The current wording blurs this distinction, i.e. the reference to lab work in this sentence implies direct methods, what follows is all about indirect popgen estimates. Although an obvious point, it would be helpful to explicitly say that indirect popgen based inference of recombination is attractive because it requires much less effort/lab-work than direct estimates and has been shown to match suprisingly well in (e.g. Booker et al 2017, Comeroon 2012).
 - It would help to refer to $\rho = 4 N_e r$ as the scaled recombination rate (rather than the recombination rate)
 - p1. l. 6 "indicator of the species' reproduction regime" -> reproductive mode
 - "biodiversity characterisation" and "breeding strategies" It was not clear to me what was meant by these terms.
 - p1. l. 40 "While it makes simplifying assumptions about the recombination process, SMC-based inference relies on the recombination rate..." it help to say in which direction we expect SMC based estimates of ρ to be biased here.
 - p1. l. 52 "Other approaches made use" Using a consistent tense here would improve flow
 - p. "When plotting the simulated population recombination rate, the average N_e is estimated using the formula $N_e = \pi / (4 \cdot u)$, where π is the average pairwise heterozygosity of the simulated dataset (computed using the vcfTools (Danecek et al. 2011)) and $u = 1.25e - 8$ is the mutation rate." It would be worth explaining this more clearly. As I understand it, the true vs simulated plots are showing $r \cdot \pi / u$ when π is affected by demography, yes?
 - p. 4, l1 "Convergence issues...." are these likely to be identifiability issues?
 - p. 4 l. 10 "Enforcing a model with flat demography using 30 intervals and a single population size only improved the inference of ρ when the recombination was zero" I did not understand why/how the number of intervals can matter when N_e is forced to be the same? Perhaps this cab be explained.
 - p. 4 l. 53 "In the "growth" scenario, a population of 50,000 increases exponentially to a size of 200,000 starting 1,000 generations ago. While population growth had little impact on the results" This seems entirely unsurprising, given that the assumed growth only affects a very recent period during which the coalescence is unlikely (for small n) even if N_e had remained constant. Why was this growth scenario chosen?
 - p. 102 "While AIC seems able to disentangle the two models, the threshold to use is ambiguous, meaning that simulations tailored for the data set to analyse are required to make a proper goodness-of-fit test. Similar conclusions are reached when considering the LRT, where lower P-values are obtained on heterogeneous data sets, but P-values lower than 1%, and even 0.01%, are observed with homogeneous data sets." Why should one use the AIC at all, if simulations are required to find meaningful threshold under M_0 ? Given that the AIC is just a penalisation of $\Delta \ln L$, would one not obtain exactly the same

results using a simulation based thresholds for LRT and AIC?

- Fig 2. and the statement "The maximum likelihood estimator appeared to be biased under this scenario, while the posterior average estimate recovered the simulated recombination rate, providing the rate was not zero". I am puzzled by the difference in bias between the MLE of the mean ρ and posterior average of ρ SMC estimates for two reasons: 1) In what way can the mean ρ be different from the average of the variable ρ ? 2) In all figures apart from Supplementary Figure 5, the results for these two appear identical.
- Fig 3. the caption is referring to circles, but no circles appear in the Figure.
- Results subheadings all start with "inference under... " which is rather repetitive.
- p. 8 "As SMC-based methods and heRho permit the inference of recombination rates from very small samples, it is possible to pre-filter the data to obtain homogeneous sub-samples that show no or little population structure."
I suggest to remove this given that subsampling of diploid individuals is unlikely to alleviate the problem: population structure changes the joint distribution of pairwise coalescence times even in a single diploid sample. One may be able to remove the effect of structure by subsampling a single haplotype per deme, but I think this is not what is meant here.

Minor fixes:

p. 1. l 32 "populations' demography" -> demography history

p. 2. l. 92 "a non-null" -> a non-zero

p. 3, l. 65 "to evaluate the robustness of recombination inference to deviations, be them simple, from a constant population size."

The grammar does not work here:

-> to evaluate the robustness of recombination inference to deviations from the assumption of a constant population size."

p.4, l. 47 "simple demographics, featuring exponential" -> simple demographic models: exponential growth or decline

p. 5, l.2 forgot closing brackets

p. 5 l. 30 the iSMC simpler model -> the simpler iSMC model

p. 5, l. 46 "We note that all methods achieve similar performance..." this is ambiguous similar to what: performance under larger N_e scenario or similar to one another?!

p. 5. l. 50 "the non-accounting of the history..." -> not accounting for the history...

p. 5, l. 56 "Noteworthy"
sounds odd/wrong in English

p. 5, l. 57 "ten fold decrease"
this contradicts the description of the decline scenario on page 2 (from 100,000 to 1,000)

p. 6, l 1 "the heterozygosity-based inference underestimates" This is referring to ρ SMC, yes?

p. 7 "null and non-null rate" -> zero and non-zero

p. 9, l1 "... although only when the recombination track is short"
should this read " although only when the mean GC track length is short"?

Dear Genetics editors and reviewers,

Thank you very much for the positive feedback and suggestions to further improve the manuscript. Please find hereby an edited manuscript incorporating all suggestions, as well as a detailed answer to each point raised.

Best regards,

Julien Dutheil.

Editor's comments:

Most important are: i) a careful revision of the introduction to better motivate the study, ii) a general fine-combing to condense the writing and fix english grammar issues (see my comments below),

Answer: The introduction has been revised as suggested and the whole manuscript as been checked to condense the writing where possible. The resulting manuscript is, however, not significantly shorter, owing to the additional clarifications in some parts.

iii) addressing the concern of reviewer 2 regarding the prevalence of GC. The fact that demographic inference appears very sensitive to plausible rates of GC is an important result that should be highlighted, vi) checking Figure 3 matches the caption v) revising subheadings.

Answer: The manuscript has been edited as suggested (see answer's to Reviewer's 2 comments for details).

Reviewer comments:

Reviewer #2 (Comments for the Authors (Required)):

In response to comment 4, the author said:

"The demography itself, however, is properly inferred if gene conversion remains below 50% of recombination events"

However, as far as I know, gene conversion is believed to be more prevalent than crossovers. Mechanistically, both crossovers and gene conversion are induced firstly from double strand break, then from the break it is more likely to initiate a conversion than a crossover. So it is slightly concerning to see that the demography inference only works for low-conversion rates. A

more meaningful benchmark with more realistic parameters might be needed here. However, please do correct me if I interpret the scenario wrong.

Answer: In mammals and plants at least, the proportion of GC events compared to CO events is quite large. The GC track length, however, is thought to be generally small (e.g. <https://www.nature.com/articles/nrg2712>), so that most of GC events will leave no trace in the population.

I found that MSMC's inference was unaffected with less than 50% of gene conversion events. With higher frequency, the effect of GC is manifest by an increased variance in the inference. Things start to really go wrong with GC > 90%, which probably reflects a lack of CO events rather than a confounding effect of GC itself. The corresponding results description was rephrased to be more precise:

“MSMC's demography inference is largely unaffected under 50% of GC events and is still accurate but with increased variance up to 90% GC events (Supplementary Figure 12). Whether this effect results from GC itself or the reduced amount of CO events would deserve further investigation.”

The other thing is that when discussing the effects of gene conversion, it remains unclear to me that how gene conversion tract length comes into play. Two simulation with the same conversion rate but different tract length will definitely show different behaviors. I would like to see some clarifications on this, and maybe the gene conversion rate has to be normalized by tract length to be comparable when tract length are different? What is the most reasonable way of combining conversion rate and tract length here to describe its net effects?

Answer: It seems to me that the track length should be compared to the heterozygosity. GC tracks that do not encompass heterozygous positions have virtually no effect. The larger the heterozygosity, the stronger the effect of GC as even with smaller track lengths will become “visible”. This is from intuition though; one would need to set up a dedicated simulation to test this properly. To fully assess the effect of GC, one should probably fix the amount of CO events and add variable amounts of GC events. In the parametrization used in this work, increasing proportions of GC events go with a lower proportion of CO events. The impact on demography inference most likely results from the reduced signal stemming from the low CO rate, and dedicated simulations will be needed to establish this with more details.

Associate Editor comments:

- *The intro should be revised carefully to motivate the study a bit better:*
 - *It seems important to say that the mean recombination rate varies widely across taxa to motivate the question of how best to infer it. It may be useful to include a sentence about recombination rate variation across taxa when citing Stapely et al 2017.*

Answer: Edited as suggested. The start of the introduction now reads as:

‘Recombination is a fundamental process impacting the segregation of alleles in populations and, consequently, a driver of genetic diversity between populations and within genomes. The average recombination rate was shown to vary extensively between individuals, populations, and species (Stapley et al. 2017).’

◦ *l. 13 "... involving various degrees of laboratory work (Peñalba and Wolf 2020)." Please distinguish more clearly between direct and indirect methods for inferring recombination rates here (and highlight the different timescales involved). The current wording blurs this distinction, i.e. the reference to lab work in this sentence implies direct methods, what follows is all about indirect popgen estimates. Although an obvious point, it would be helpful to explicitly say that indirect popgen based inference of recombination is attractive because it requires much less effort/lab-work than direct estimates and has been shown to match surprisingly well in (e.g. Booker et al 2017, Comeroon 2012).*

Answer: This paragraph was rephrased as:

“Recombination rates can be inferred either directly or indirectly (Peñalba and Wolf 2020). Direct assessments provide measures of the contemporary molecular recombination rate, and involve experimentally demanding methods such as multiple crossing and genotyping. Indirect methods involve genome resequencing in combination with variant calling pipelines, and (i) require comparatively less laboratory work, (ii) are also amenable to organisms that cannot be grown in the laboratory. These methods rely on the patterns that recombination leaves on the distribution of variants within populations; they infer the so-called population recombination rate, or scaled recombination rate, noted ρ , equal to $2 \cdot x \cdot N_e \cdot r$, where N_e is the effective population size, x the ploidy of the individuals serving as unit count for N_e , and r is the “molecular” recombination rate, the number of recombination events per base pair per generation. As opposed to direct methods, indirect methods provide a historical recombination rate, averaged over multiple individuals and generations.”

◦ *It would help to refer to $\rho = 4 N_e r$ as the scaled recombination rate (rather than the recombination rate)*

Answer: I have addressed this suggestion in two ways: first, I referred to ρ as the “population recombination rate” (term that I personally prefer to “scaled” recombination rate), and I have replaced multiple occurrences of the term “recombination rate” by just “ ρ ”, which has simplified the text.

◦ *p1. l. 6 "indicator of the species' reproduction regime" -> reproductive mode*

Answer: Corrected as suggested.

◦ *"biodiversity characterisation" and "breeding strategies" It was not clear to me what was meant by these terms.*

Answer: This paragraph has been rephrased with the hope to make it clearer. It now reads as:

“The amount of recombination occurring in a population is an indicator of the species’ reproduction mode, providing insights into the frequency of sexual reproduction in natural populations (e.g. in plankton species Rengefors et al. (2017)). The recombination rate also determines how efficiently populations get rid of deleterious mutations and fix advantageous ones, with implications for conservation biology (Theissinger et al. 2023) and breeding strategies (Epstein et al. 2023)”

◦ p1. l. 40 *"While it makes simplifying assumptions about the recombination process, SMC-based inference relies on the recombination rate..." it help to say in which direction we expect SMC based estimates of rho to be biased here.*

Answer: The corresponding section has been rephrased and extended. It now reads as:
“The SMC is an approximation of the sequential coalescent with recombination (Wu and Heine 1999). SMC models have a recombination rate parameter, which can be jointly estimated with demographic parameters in inference methods like PSMC (Li and Durbin 2011) and MSMC2 (Schiffels and Wang 2020). Because they ignore certain classes of recombination events (Marjoram and Wall 2006), SMC models are expected to underestimate rho. Furthermore, most SMC models use a single parameter for the entire analysed genome, an assumption at odds with many empirical recombination landscapes, which display skewed rate distributions. Conversely, the iSMC method accounts for the heterogeneity of the recombination rate by assuming a prior distribution and an auto-correlation process along the genome (Barroso et al. 2019). In this model, the genome-wide recombination rate is estimated as a hyper-parameter (the mean of the prior distribution).”

◦ p1. l. 52 *"Other approaches made use" Using a consistent tense here would improve flow*

Answer: Corrected.

• p. *"When plotting the simulated population recombination rate, the average Ne is estimated using the formula $N_e = \pi / (4 \cdot u)$, where π is the average pairwise heterozygosity of the simulated dataset (computed using the vcftools (Danecek et al. 2011)) and $u = 1.25e - 8$ is the mutation rate." It would be worth explaining this more clearly. As I understand it, the true vs simulated plots are showing $r \cdot \pi / u$ when π is affected by demography, yes?*

Answer: This is correct. The alternative would have been to plot r. However, estimating r requires knowledge of the mutation rate, which is not the case in most real cases. I chose to plot rho which is the quantity actually estimated by most methods. The corresponding text has been rephrased as:

“Importantly, under a non-constant population size scenario, $\rho = 4 \cdot N_e \cdot r$ varies in time, even if r is constant. The inferred rho is, therefore, a time average of the population recombination rate. When plotting the simulated population rho, in the absence of selection, the average Ne is estimated using the formula $N_e = \pi / (4 \cdot u)$, where pi is the average pairwise heterozygosity of the simulated dataset (computed using the vcftools (Danecek 2011)) and u = 1.25e-8 is the mutation rate.”

- p. 4, l1 *"Convergence issues.... " are these likely to be identifiability issues?*

Answer: I do not think so, as a larger dataset (longer genome sequence) or more diverse (higher mutation rate) would most certainly permit the estimation of these parameters.

- p. 4 l. 10 *"Enforcing a model with flat demography using 30 intervals and a single population size only improved the inference of ρ when the recombination was zero" I did not understand why/how the number of intervals can matter when N_e is forced to be the same? Perhaps this can be explained.*

Answer: The number of intervals determines the number of states in the HMM, that is, how finetuned the discretization is. In the original PSMC and several extensions, the demographic model and the discretization are largely confounded, using roughly one discrete category per time interval (but that is not the case of SMC++ and iSMC, which use splines for the demographic models while discretizing time for the HMM). Reducing the number of categories is then a way to reduce the number of parameters. MSMC2 allows the user to play with this parametrization, allowing to reduce the number of estimated parameters while leaving the HMM state space unchanged. I used this possibility to assess the effect of the number of parameters independently of the discretization scheme, in response to one of the reviewer's comments. I have further updated the manuscript with the hope to make this part clearer, it now reads as:

"To disentangle the effect of discretization (impacting the number of hidden states in the hidden Markov chain) and of the number of parameters to estimate, a model with flat demography using 30 intervals and a single population size parameter was fitted. Such a model only improved the inference of ρ when the simulated recombination rate was zero (Supplementary Figure 2)."

- p. 4 l. 53 *"In the "growth" scenario, a population of 50,000 increases exponentially to a size of 200,000 starting 1,000 generations ago. While population growth had little impact on the results" This seems entirely unsurprising, given that the assumed growth only affects a very recent period during which the coalescence is unlikely (for small n) even if N_e had remained constant. Why was this growth scenario chosen?*

Answer: The growth scenario was chosen as the "symmetric" of the decline scenario, with some population size adjustment to avoid a too low genetic diversity. I do agree this was not such a good choice, though, as the distribution of coalescence events is very different in both cases. I have run additional analyses with a growth starting from 10,000 individuals 100k generations ago, to a population size of 200,000 at present. All figures have been updated to include both scenarios (now referred to as "recent growth" and "ancient growth"). The new, ancient population growth scenario has an impact on the inference (like population decline has), and the corresponding results have been updated accordingly.

- p. 102 *"While AIC seems able to disentangle the two models, the threshold to use is ambiguous, meaning that simulations tailored for the data set to analyse are required to make a proper goodness-of-fit test. Similar conclusions are reached when considering the LRT, where lower P-values are obtained on heterogeneous data sets, but P-values lower than 1%, and even 0.01%, are observed with homogeneous data sets." Why should one use the AIC at all, if simulations are required to find meaningful threshold under MO? Given that the AIC is just a penalisation of $\Delta \ln L$, would one not obtain exactly the same results using a simulation based thresholds for LRT and AIC?*

Answer: This is a good point. The following conclusion has been added to this section: "AIC and LRT model comparison tests do not permit to properly distinguish SMC models with heterogeneous and homogeneous recombination. Tailored simulations are required to derive significance likelihood thresholds."

- *Fig 2. and the statement "The maximum likelihood estimator appeared to be biased under this scenario, while the posterior average estimate recovered the simulated recombination rate, providing the rate was not zero". I am puzzled by the difference in bias between the MLE of the mean rho and posterior average of rhoSMC estimates for two reasons: 1) In what way can the mean rho be different from the average of the variable rho? 2) In all figures apart from Supplementary Figure 5, the results for these two appear identical.*

Answer: The genome-averaged recombination rate is a (hyper)parameter of the iSMC prior gamma distribution of rates. It is estimated together with demography parameters using a Maximum Likelihood procedure. It is also possible to get a posterior average by computing the mean of the posterior distribution of rates. In simulations where the recombination rate distribution is close to the prior distribution, both estimates tend to be highly similar. When the real distribution departs from the prior gamma distribution (for instance when it is flat), the ML estimates is biased, but the posterior can better capture the shape of the real distribution and provide a better estimate. The corresponding Methods section has been edited with the hope to make this aspect clearer:

"We further considered a model with a five-class discrete gamma model of recombination, which we here note rhoSMC. Under this model, the genome-average recombination rate ρ is the mean of the gamma prior distribution of rates and is estimated together with other parameters using a maximum likelihood approach. The rhoSMC model further permits to use an empirical Bayesian approach to estimate the site-specific recombination by computing the mean of each recombination class weighted by their posterior probabilities, computed using the backward algorithm (Dutheil 2021). A posterior estimate of the genome-average population recombination rate is then computed as the mean over all site-specific posterior estimates.

- *Fig 3. the caption is referring to circles, but no circles appear in the Figure.*

Answer: thank you for noticing this error! The Figure and corresponding script have been corrected.

- *Results subheadings all start with "inference under..." which is rather repetitive.*

Answer: Subheadings have been updated and simplified. These now read as:

Ideal scenario: flat recombination landscape and constant population size

Non-constant population size

Heterogeneous recombination landscape

Recombination hotspots

Gene conversion

Testing for the occurrence of recombination

- *p. 8 "As SMC-based methods and $heRho$ permit the inference of recombination rates from very small samples, it is possible to pre-filter the data to obtain homogeneous sub-samples that show no or little population structure."*

I suggest to remove this given that subsampling of diploid individuals is unlikely to alleviate the problem: population structure changes the joint distribution of pairwise coalescence times even in a single diploid sample. One may be able to remove the effect of structure by subsampling a single haplotype per deme, but I think this is not what is meant here.

Answer: This paragraph was simplified and now reads as

"Population structure and admixture events were reported to bias LD-based estimates of recombination rates (Samuk and Noor 2022); it will be important to assess how they affect the inference of recombination rates with $heRho$ and SMC-based methods in future studies."

Minor fixes:

p. 1. l 32 "populations' demography" -> demography history

Answer: changed to "demographic history"

p. 2. l. 92 "a non-null" -> a non-zero

Answer: changed as suggested (also other places in the manuscript)

p. 3, l. 65 "to evaluate the robustness of recombination inference to deviations, be them simple, from a constant population size."

The grammar does not work here:

-> to evaluate the robustness of recombination inference to deviations from the assumption of a constant population size."

Answer: changed as suggested.

p.4, l. 47 "simple demographics, featuring exponential" -> simple demographic models: exponential growth or decline

Answer: changed as suggested.

p. 5, l.2 forgot closing brackets

Answer: fixed.

p. 5 l. 30 the iSMC simpler model -> the simpler iSMC model

Answer: changed as suggested.

p. 5, l. 46 "We note that all methods achieve similar performance... " this is ambiguous similar to what: performance under larger N_e scenario or similar to one another?!

Answer: now rephrased as "We note that the methods' performances were unaffected when simulating under a population size ten times smaller"

p. 5. l. 50 "the non-accounting of the history..." -> not accounting for the history...

Answer: changed as suggested.

*p. 5, l. 56 "Noteworthy"
sounds odd/wrong in English*

Answer: changed to "Notably".

*p. 5, l. 57 "ten fold decrease"
this contradicts the description of the decline scenario on page 2 (from 100,000 to 1,000)*

Answer: this was indeed a mistake, and has been corrected to "hundred-fold decrease".

p. 6, l 1 "the heterozygosity-based inference underestimates" This is referring to rhoSMC, yes?

Answer: this refers to the heRho method. The text has been updated to make it explicit.

p. 7 "null and non-null rate" -> zero and non-zero

Answer: corrected as suggested.

*p. 9, l1 "... although only when the recombination track is short"
should this read " although only when the mean GC track length is short"?*

Answer: indeed, this is now corrected.

March 14, 2024

RE: GENETICS-2024-306814R1

Prof. Julien Y Dutheil
Max-Planck-Institut für Evolutionsbiologie
Theoretical Biology
August Thienemann Strasse 2
Plön 24306
Germany

Dear Dr. Dutheil:

Congratulations! We are delighted to inform you that your manuscript entitled "On the estimation of genome-average recombination rates" is acceptable for publication in GENETICS. Many thanks for submitting your research to the journal.

To Proceed to Production:

1. Format your article according to GENETICS style, as discussed at <https://academic.oup.com/genetics/pages/general-instructions>, and upload your final files at <https://genetics.msubmit.net>.
2. Your manuscript will be published as-is (unedited-as submitted, reviewed, and accepted) at the GENETICS website as an Advanced Access article and deposited into PubMed shortly after receipt of source files and the completed license to publish. Please notify sourcefiles@thegsajournals.org if you do not wish to publish your article via Advanced Access.
3. We invite you to submit an original color figure related to your paper for consideration as cover art. Please email your submission to the editorial office or upload it with your final files. You can submit a small-sized image for evaluation, and if selected, the final image must be a TIFF file 2513px wide by 3263px high (8.375 by 10.875 inches; resolution of 600ppi). Please avoid graphs and small type.

If you have any questions or encounter any problems while uploading your accepted manuscript files, please email the editorial office at sourcefiles@thegsajournals.org.

Sincerely,

Konrad Lohse
Associate Editor
GENETICS

Approved by:
Nicholas Barton
Senior Editor
GENETICS

note: Please add jnls.author.support@oup.com and genetics.oup@kwglobal.com (or the domains @oup.com and @kwglobal.com) to your email program's "safe senders" list. You will be contacted by both at various points during the production process.